# SmartChunk Retrieval: Query-Aware Chunk Compression with Planning for Efficient Document RAG

**Xuechen Zhang**[*]
University of Michigan
zxuechen@umich.edu

**Koustava Goswami**
Adobe Research
koustavag@adobe.com

**Samet Oymak**
University of Michigan
oymak@umich.edu

**Jiasi Chen**
University of Michigan
jiasi@umich.edu

**Nedim Lipka**
Adobe Research
lipka@adobe.com

## Abstract

Retrieval-augmented generation (RAG) has strong potential for producing accurate and factual outputs by combining language models (LMs) with evidence retrieved from large text corpora. However, current pipelines are limited by static chunking and flat retrieval: documents are split into short, predetermined, fixed-size chunks, embeddings are retrieved uniformly, and generation relies on whatever chunks are returned. This design brings challenges, as retrieval quality is highly sensitive to chunk size, often introduces noise from irrelevant or misleading chunks, and scales poorly to large corpora. We present SmartChunk retrieval, a query-adaptive framework for efficient and robust long-document question answering (QA). SmartChunk uses (i) a planner that predicts the optimal chunk abstraction level for each query, and (ii) a lightweight compression module that produces high-level chunk embeddings without repeated summarization. By adapting retrieval granularity on the fly, SmartChunk balances accuracy with efficiency and avoids the drawbacks of fixed strategies. Notably, our planner can *reason about* chunk abstractions through a novel reinforcement learning scheme, STITCH, which boosts accuracy and generalization. To reflect real-world applications, where users face diverse document types and query styles, we evaluate SmartChunk on five QA benchmarks plus one out-of-domain dataset. Across these evaluations, SmartChunk outperforms state-of-the-art RAG baselines, while reducing cost. Further analysis demonstrates strong scalability with larger corpora and consistent gains on out-of-domain datasets, highlighting its effectiveness as a general framework for adaptive retrieval.

## 1 Introduction

Large language models (LLMs) have achieved remarkable progress in recent years, demonstrating strong performance across a wide range of natural language processing tasks, particularly question answering (QA) (Team et al. (2023); Grattafiori et al. (2024)). QA is becoming an integral part of different assistance tools (e.g., ChatGPT, Copilot) which users increasingly rely on for information seeking, decision support, and task completion. While LLMs can act as effective knowledge stores and can be further adapted through fine-tuning, they unavoidably struggle with knowledge that changes over time or lies outside their pretraining distribution (Xu et al. (2024); Zhang et al. (2023); Huang et al. (2025); Kandpal et al. (2023)). To address these limitations, retrieval-augmented generation (RAG) has emerged to enhance generation accuracy in knowledge-intensive tasks by retrieving relevant information from external corpora and providing it to the LLM as additional context (Chen et al. (2017); Ram et al. (2023); Akyürek et al. (2022); Izacard & Grave (2020)). Despite strong promise, current RAG pipelines break down for long-document QA. Imperfect retrievers introduce

---

[*]Work done during an internship at Adobe.

noisy or even misleading evidence (Su et al. (2024); Yoran et al. (2023); Khattab et al. (2023); Shi et al. (2023)), and generators (LLMs) struggle to process very long inputs (An et al. (2024); Liu et al. (2023); Xu et al. (2023); Li et al. (2024b)). As a result, answer quality often degrades as the corpus size increases.

In practice, most multi-document systems follow a static pipeline: documents are split into short, fixed-size chunks, embeddings are retrieved independently with each treated equally (i.e., non-hierarchical retrieval), and the top-$k$ chunks are then passed to the LLM. Controlled experiments indicate that this static chunking–retrieval design is the primary bottleneck as retrieval quality is highly sensitive to chunk size, and no single granularity works well across heterogeneous queries and document structures (Bhat et al. (2025)). Recent work attempts to improve chunking via recursive chunking, sliding windows, paragraph-based splits, semantic boundaries or contextual chunking (LangChain team (2023); Kamradt (2024); Günther et al. (2024); Anthropic

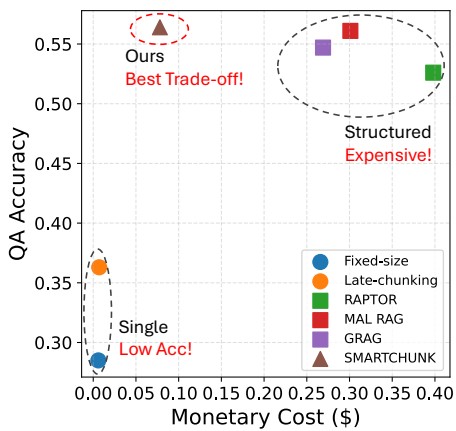

Figure 1: QA accuracy vs. Monetary cost across methods. SMARTCHUNK achieves higher accuracy with lower cost compared to state-of-the-art baselines.

(2024)). While these methods offer some improvements, they remain constrained by static pre-processing pipelines, which cannot adapt to diverse queries and often suffer from noisy retrieval or "lost-in-the-middle" effects (Liu et al. (2023)). More recently, tree- and graph-structured RAG methods (Sarthi et al. (2024); Zheng et al. (2025); Chen et al. (2023); Zhang et al. (2025a); Edge et al. (2024); Liang et al. (2024); Sun et al. (2023a); Li et al. (2024a); Liu et al. (2025); Fazili et al. (2024); Nair et al. (2023)) organize text into hierarchical structures to support multi-hop reasoning, but these systems introduce substantial complexity, computational cost, and dependence on structured representations. Other works also address these issues implicitly by finetuning LMs to predict correct answers despite noisy retrieval. However, such finetuning is computationally expensive, time-consuming, and often reduces interpretability (Chen et al. (2024); Cuconasu et al. (2024); Liu et al. (2023); Wu et al. (2024)). These limitations highlight the need for lighter-weight, query-adaptive chunking frameworks that can dynamically adjust abstraction levels while remaining efficient.

In this paper, we present SMARTCHUNK, a query-aware framework for dynamic chunk generation and retrieval in long-document QA. Our approach addresses both efficiency and adaptability through three key innovations:

- **Planner for Query-Aware Chunking.** We introduce a lightweight planner that decides the smallest and largest chunk sizes required to answer a query, enabling multi-level retrieval that adapts to both query complexity and document structure, reducing unnecessary chunk expansions. The planner is designed to be accurate, low-latency ($\leq$ 1s), and adaptable across diverse domains and document types.

- **STITCH for Robust Planner Training.** Training the planner is challenging because ground-truth labels are unavailable, supervised traces are costly to generate, and pseudo-labels are often noisy. In addition, multi-objective reinforcement learning(RL), balancing accuracy, monetary cost, and latency, can be unstable. We propose *STITCH (Solve with RL, Then Imitate To Close Holes), a stable RL ↔ SFT loop* that alternates between RL, hinted RL, and imitation learning. To enable this process, we also design a pseudo-labeling and reasoning trace generation pipeline. This framework enables stable multi-objective training with improved sample and computational efficiency, delivers strong planning performance.

- **Compressor for Compact Representations.** We introduce a lightweight compression module that produces embeddings capturing the high-level semantics of large document spans, without invoking expensive summarizers based on large LLMs (e.g., GPT). This addresses a key limitation of structured chunking: while building hierarchical chunks can be effective, it is very expensive. By providing coarse compressed views alongside fine-grained local context, the compressor

achieves the benefits of structured chunking more efficiently, balancing retrieval accuracy with token and computation cost.

**Strong Empirical Results.** Across four QA benchmarks and one out-of-domain dataset, SMARTCHUNK outperforms state-of-the-art RAG baselines while reducing monetary cost(each query incurs access fees from LLM APIs, often a few cents per call) by 30%, as shown in Figure 1. Extensive ablation studies validate the efficiency of our design choices, and detailed evaluations show that SMARTCHUNK achieves the best trade-off between cost (including monetary cost and latency) and accuracy. Furthermore, we demonstrate that SMARTCHUNK is orthogonal to other advances in RAG pipelines, such as late chunking( Günther et al. (2024)) and Hybrid Search (Liu & contributors (2022)). When combined, these methods show even greater improvements.

## 2 RELATED WORK

**Chunking Optimization** Chunking strategy plays a central role in retrieval-augmented generation (RAG). While early approaches use fixed-length or sentence-level chunking (Lewis et al. (2020)), recent work shows that static heuristics are often suboptimal (Bhat et al. (2025)). To address the challenge of missing or fragmented context, several methods augment chunking strategies, such as recursive chunking, sliding window chunking, paragraph-based chunking, and semantic chunking (LangChain team (2023); Kamradt (2024); Günther et al. (2024)). Anthropic's Contextual Retrieval (Anthropic (2024)) proposes generating contextualized chunk embeddings using a secondary LLM, improving semantic fidelity at the cost of greater computational overhead. However, these techniques rely on static preprocessing, which limits their ability to adapt to varying query complexity and can lead to noisy retrievals or the "lost-in-the-middle" issue (Liu et al. (2023)).

**Tree & Graph Structured Retrieval Augmented Generation** Recent work has explored tree and graph structures as alternatives to vanilla RAG for handling large, noisy corpora and complex reasoning. Tree-structured methods such as RAPTOR (Sarthi et al. (2024)), MAL (Zheng et al. (2025)) and MemWalker (Chen et al. (2023)) recursively embed, cluster, and summarize passages into multi-level trees. SiReRAG (Zhang et al. (2025a)) constructs both similarity and relatedness trees. PG-RAG (Liang et al. (2024)), GraphRAG (Edge et al. (2024)) HopRAG (Liu et al. (2025)) DALK (Li et al. (2024a)) and ToG (Sun et al. (2023a)) prompts LLMs to organize knowledge into mindmaps. While tree & graph-based RAG effectively models hierarchical logic, they often incur higher cost, complexity, and reliance on structure rather than plain text. These trade-offs motivate lighter-weight, query-adaptive alternatives that can flexibly combine multi-level abstraction with efficiency.

**Finetuning efficient and controllable SLMs** There is growing evidence that small language models (SLMs) can power practical agents when finetuned with the right signals and constraints (Belcak et al. (2025)). Recent works focus on methods that can make LLMs effective reasoners. Error-driven approaches such as self-improvement loops (Shinn et al. (2023)) show good performance. With the emergence of (Guo et al. (2025); Shao et al. (2024); Team et al. (2025); Qin & Springenberg (2025)), more and more reinforcement finetuned reasoning models demonstrate the importance of RL (Yu et al. (2025); Lin et al. (2025); Xu et al. (2025)) in reasoning tasks. Distillation works (Muennighoff et al. (2025); Guo et al. (2025)) prove that SFT can help SLMs acquire reasoning capability comparable to the expert/teacher models. However, reliably finetuning SLMs for reasoning, especially if we still want to keep cost/latency low, remains challenging (Zhang et al. (2025b;c); Li et al. (2025); Qu et al. (2025); Ma et al. (2025); Chu et al. (2025)). Pursuing controllability (e.g., constraining reasoning length) also remains a difficult problem: recent work explores objective design (Zhang et al. (2025c); Aggarwal & Welleck (2025); Shrivastava et al. (2025)), but it often requires high computational cost and complex training pipelines. In contrast, we propose STITCH, which achieves both strong performance and concise reasoning traces in a more efficient manner.

## 3 PROBLEM STATEMENT

We study the RAG problem, where the goal is to provide accurate responses to user queries from documents. Our focus is on long-document QA, a particularly challenging task due to variation in document length, domain, and structure, as well as diversity in query types ranging from abstract to detailed and multi-hop.

The problem is to answer a user query $q \in \mathcal{Q}$, where the correct answer $\hat{a}$ depends on retrieving a subset of useful chunks $\hat{C} \subseteq D$, where $D$ is the reference corpus segmented into chunks $C$.

Figure 2: Left: Overview of the SMARTCHUNK framework. Compared to vanilla RAG, which uses fixed chunking and flat retrieval, SmartChunk introduces two key modules: (1) a planner $\mathcal{P}$ that predicts the smallest and largest chunk sizes per query, enabling adaptive multi-level retrieval, and (2) a Chunk Compression Encoder $\mathcal{E}$ that produces compact, high-level embeddings for aggregated chunks, lowering the cost of the multi-level representation. These additions allow SMARTCHUNK to adapt to different query complexity and document structure, balancing accuracy and efficiency. Modules added by SmartChunk are shown in blue, while modules from vanilla RAG are shown in black. The figure distinguishes between text (represented by blocks with horizontal lines) and embeddings (shown as solid-colored blocks). Right: The STITCH method for Planner training.

The retrieval task $\pi$ is to identify all and only the useful chunks $\hat{C}$ from $C$, and supply them to a downstream generator $\mathcal{G}$, typically a large language model (LLM) such as GPT-4o Hurst et al. (2024), to produce the final response $a = \mathcal{G}(q, \hat{C})$. The key difficulty lies in retrieving the right chunks from large corpora while avoiding noise, fragmentation, and excessive token usage, so that the generator can produce an accurate answer.

The goal is to maximize response accuracy while minimizing the costs associated with retrieval and generation. We formulate this as a joint optimization problem that balances two objectives:

- Answer accuracy: probability that $a = \mathcal{G}(q, \pi(q, D))$ produces the correct answer $\hat{a}$.
- Cost Efficiency: minimizing the overall cost $\tau(\pi, \mathcal{G})$, which includes the token budget (correlating to monetary cost) and the latency incurred by embedding, retrieval, and generation.

The goal of our framework is to learn a policy $\pi$ that optimizes the trade-off between accuracy and cost efficiency: $\max_\pi \ \mathbb{E}_{(q,D,\hat{a}) \sim \mathcal{D}} \Big[ \text{Acc}(\mathcal{G}(q, \pi(q, C))), \hat{a}) - \lambda \tau((\mathcal{G}, \pi \mid q, D)) \Big]$, where $\lambda > 0$ controls the trade-off between accuracy and efficiency. We consider two types of costs in this work, monetary price and latency. The details are shown in Section 5.1.

# 4 PROPOSED FRAMEWORK: SMARTCHUNK RETRIEVAL

## 4.1 OVERVIEW OF SMARTCHUNK RETRIEVAL

We propose the SMARTCHUNK framework, depicted in Figure 2. When a new question $q \in \mathcal{Q}$ arrives, and there is a reference corpus of documents $D$ that is segmented into a set of chunks $C$, the planner $\mathcal{P}$ predicts the smallest and largest chunk sizes to consider, using both document metadata and the query. In parallel, the Chunk Compression Encoder $\mathcal{E}$ builds a multi-level hierarchy by clustering smaller chunks and aggregating them into higher-level chunks. A high-level chunk embedding contains representation of the text summary of that cluster. The Chunk Compression Encoder $\mathcal{E}$ produces multi-level embeddings directly. Each high-level chunk thus encodes the semantics of a cluster without requiring explicit text summarization. This design eliminates repeated summary generation and enables efficient construction of embeddings for high-level chunks in the hierarchy. The retriever $\mathcal{R}$ and generator $\mathcal{G}$ follow the vanilla RAG system (Liu & contributors (2022)).

**Multi-level Chunking** We assume a multi-level chunk hierarchy $\mathcal{H}(D)$, where a document is represented at multiple granularities (e.g., sentence, paragraph, section, document or token spans such as 128, 256, 512, 1024 tokens) in a hierarchical structure. Higher-level chunks are constructed by aggregating lower-level ones through a summarizer $\mathcal{S}(\cdot)$. In practice, LLMs such as GPT can serve as $\mathcal{S}$ to generate summarized text that forms a higher-level chunk. While raw fine-grained chunks preserve full semantic fidelity, they may fragment higher-level structure. Conversely, compressed chunks capture compact semantic summaries but may lose fine-grained details. This design enables the system to balance accuracy versus efficiency by adaptively mixing raw and compressed representations. We provide an example of the Multi-level chunk hierarchy in Section H.

**Planner** The planner, denoted by $\mathcal{P}$, is a model that predicts the appropriate chunking granularity for a given query. Formally, $\mathcal{P}(q, MetaData(D)) = (\text{level}_{\min}, \text{level}_{\max})$, where $\text{level}_{\min}, \text{level}_{\max}$ denote the smallest and largest chunk levels sufficient to answer $q$ while minimizing overall cost (including multi-level chunk construction, retrieval, and generation), and $MetaData(D)$ denotes the metadata of documents. These predictions restrict the candidate retrieval set to $C = \{c \in \mathcal{H}(D) \mid \text{level}(c) \in [\text{level}_{\min}, \text{level}_{\max}]\}$. We provide examples of the planner's inputs and outputs in Section F.

**Chunk Compression Encoder** Beyond text-level summarization, we introduce an embedding-level compression module, denoted as $\mathcal{E}(\cdot)$. Unlike $\mathcal{S}(\cdot)$, which produces summarized text chunks, $\mathcal{E}(\cdot)$ directly maps a set of fine-grained chunks into a single compressed embedding without generating intermediate text. Formally, given a group of lower-level chunks $\{c_1, c_2, \ldots, c_4\}$, the module outputs $e(c_5) = \mathcal{E}(c_1, c_2, \ldots, c_4) \in \mathbb{R}^d$. where $e(c_5)$ is a high-level semantic embedding that captures the aggregate meaning of the input chunks. In practice, $\mathcal{E}$ can be a pretrained embedding model such as SBERT encoder (multi-qa-mpnet-base-cos-v1 Reimers & Gurevych (2019)), long-context embedding model (jina-embeddings-v3 Sturua et al. (2024)), or a finetuned Chunk Compression Encoder.

### 4.2 PLANNER: TRAINED WITH SYNTHETIC DATA PIPELINE + STITCH

The planner must satisfy several requirements to be effective in retrieval-augmented QA. Accuracy is essential, as errors in selecting chunk levels directly propagate to retrieval and degrade downstream answer quality. The planner should also operate under low latency, with a target reasoning length of fewer than 128 tokens (roughly one second on commodity hardware), ensuring practical deployment in interactive settings. Finally, the planner must exhibit adaptability, performing robustly across diverse domains, query types, and document structures. To address the challenges of training a planner, including unavailable ground-truth, noisy pseudo-labels, costly data collection, and unstable multi-objective optimization, we propose STITCH (**S**olve with RL, **T**hen **I**mitate **T**o **C**lose **H**oles), a robust and sample-efficient training framework. As illustrated in Figure 2(right) and Algorithm 2, STITCH combines reinforcement learning (RL) and supervised fine-tuning (SFT) in a stable loop.

In SMARTCHUNK, for each question $q$ paired with the documents $D$ and answer $a$, the workflow of SMARTCHUNK is shown in proceeds as follows:

1. **Vanilla RL rollout:** In Step 1, we optimize the planner using vanilla RL. Given a task $(q, D)$, the planner generates a group of $G$ rollouts $\{o_i\}_{i=1}^G = \mathcal{P}(q, MetaData(D))$, where each rollout predicts a chunking granularity plan based on the query and document metadata. For each rollout, we extract the predicted chunk sizes $(\text{level}_{\min}, \text{level}_{\max})$ and execute the corresponding retrieval-augmented QA. If any rollout leads to a correct answer and satisfies the planning latency constraint, we mark the case as solvable and use it for policy updates.
2. **Hinted RL rollout:** If the question remains unsolved, we enter Step 2 (Hinted RL rollout). We generate an expert trace that solves the question while satisfying all constraints, and extract a short hint from this trace. This hint is appended to the question $(q, \rho)$ and used to condition a new RL rollout. If the hinted rollout succeeds, the planner is updated accordingly.
3. **Imitation learning:** However, for hard cases that remain unsolved even with hints, we move to Step 3 (imitation learning). These examples are stored and periodically used to fine-tune the model via supervised learning, using full expert traces and answers.

This training loop continues, alternating between RL and SFT, until convergence. STITCH enables stable learning under multi-objective reward, improves sample and computational efficiency, and produces a planner that generalizes across domains and document types with low latency.

**Policy updates:** Solvable cases from either stage are used to update the policy with the following objective, following GRPO Shao et al. (2024). The update balances reward, policy divergence, and planning efficiency:

$$\mathcal{J}_{\text{STITCH}}(\theta) = \mathbb{E}_{(q,\rho,a)\sim\mathcal{D}, \{o_i\}_{i=1}^G \sim \pi_{\text{old}}(\cdot|(q,\rho))}$$

$$\left[ \frac{1}{G} \sum_{i=1}^G \frac{1}{|o_i|} \sum_{t=1}^{|o_i|} \left( \min\left( r_{i,t}(\theta)\hat{A}_{i,t}, \text{clip}\left(r_{i,t}(\theta), 1-\varepsilon, 1+\varepsilon\right)\hat{A}_{i,t}\right) - \beta D_{\text{KL}}(\pi_\theta||\pi_{\text{ref}})\right) \right], \quad (1)$$

where

$$r_{i,t}(\theta) = \frac{\pi_\theta(o_{i,t}|q,\rho,o_{i<t})}{\pi_{\text{old}}(o_{i,t}|q,\rho,o_{i<t})}, \quad \hat{A}_{i,t} = \frac{R_i - \text{mean}(\{R_i\}_{i=1}^G)}{\text{std}(\{R_i\}_{i=1}^G)} \quad (2)$$

**RL Reward Components in STITCH** To train the planner effectively, we design a multi-objective reward $R$ that balances several requirements. It includes correctness of the QA answer, a penalty for excessive chunk usage, a penalty for overly long reasoning traces, and a reward for producing outputs in the desired format. We also introduce a pseudo-label alignment reward, which provides stable guidance in early training by aligning the planner's outputs with precomputed labels before it fully explores the performance–efficiency trade-off. Details of the formulations are provided in Section G.

**Synthetic data pipeline** To support this loop, we design a reasoning trace generation pipeline with four stages:

1. **Hierarchy Construction.** We first construct the complete chunk hierarchy, with the smallest unit corresponding to a sentence and the largest corresponding to the full document.
2. **Initial Retrieval.** Given a query, we perform top-k retrieval and use the retrieved chunks to generate an answer with a backbone model.
3. **Pseudo-Label Assignment.** If the generated answer is correct, we record the smallest and largest chunk levels among the retrieved chunks as a pseudo-label. We emphasize that these pseudo-labels are not guaranteed to be optimal. If the answer is incorrect, we expand retrieval with additional chunks to check whether the model can then recover the correct answer.
4. **Reasoning Trace Generation.** We use the collected pseudo-labels to generate reasoning traces. To improve trace diversity and reduce overfitting, we sample traces from diverse large language model families, covering parameter scales from 1.5B to 671B. The decision choice on the same has been covered in the Appendix.

This pipeline enables automatic construction of supervision signals that are both diverse and scalable, providing a foundation for robust planner training. We shown ablation experiments of sample traces from multiple large language models or a single LLM and the example of generated trace in Section F.

### 4.3 COMPRESSOR

A straightforward approach to obtaining high-level chunk embeddings is to first generate a summary with a LLM and then encode that summary using an embedding model. Formally, given a group of fine-grained chunks $\{c_1, \ldots, c_m\}$, we obtain a summarized text chunk via an LLM: $\hat{s} = \mathcal{S}(c_1, \ldots, c_m)$, and its embedding using a pretrained encoder $\epsilon$: $e_{\text{gt}} = \epsilon(\hat{s}) \in \mathbb{R}^d$. This provides a ground-truth compressed embedding $e_{\text{gt}}$. However, this pipeline is expensive because it requires repeated calls to the LLM $\mathcal{S}$.

To reduce cost, we train a lightweight compression model $\mathcal{S}$ that maps a set of lower-level chunk embeddings directly into a high-level compressed embedding: $e_{\text{comp}} = \mathcal{S}\big(\epsilon(c_1), \ldots, \epsilon(c_m)\big)$. The compression model is trained to minimize the discrepancy between its output $e_{\text{comp}}$ and the ground-truth embedding $e_{\text{gt}}$: $\mathcal{L}_{\text{comp}}(\mathcal{S}) = \big\|e_{\text{comp}} - e_{\text{gt}}\big\|_2^2$. Once trained, $\mathcal{S}$ can directly produce high-level embeddings without requiring LLM summarization, yielding comparable retrieval accuracy at much lower cost in both compute and latency.

We further discuss our design choices and their justification in Section I.

## 5 EXPERIMENTS

### 5.1 EXPERIMENTAL SETTING

**Datasets** We extensively evaluate the effectiveness and efficiency of SMARTCHUNK across five QA benchmarks that span diverse document types and question formats, aiming to reflect realistic retrieval-augmented generation (RAG) scenarios. The datasets are shown in Table 1. Following prior work (Sarthi et al. (2024); Bhat et al. (2025)), we include four in-domain datasets, which cover narrative comprehension, academic question answering, long-form multiple-choice QA, and structured table/text QA respectively. To assess generalization, we additionally include an out-of-distribution benchmark, NewsQA, which is grounded in CNN news articles and introduces a new domain with diverse topics and more open-ended query–document patterns, providing a challenging testbed for cross-domain robustness.

**Evaluation** We evaluate both the effectiveness and efficiency of our system using standard metrics established in prior work (Sarthi et al. (2024); Bhat et al. (2025); Zheng et al. (2025)). Our evaluation focuses on three key dimensions: retrieval quality, end-to-end answer quality, and system efficiency.

| Dataset | # Docs | Doc Length | # QA Pairs | Document Type | Query Type |
|---------|--------|-----------|-----------|---------------|-----------|
| NarrativeQA (Kočiský et al. (2018)) | 1.6k | 62k | 46k | Fictional stories | Abstractive narrative comprehension |
| QASPER (Dasigi et al. (2021)) | 1.5k | 4.1k | 5.0k | NLP papers | Extractive fact-based |
| QuALITY (Pang et al. (2021)) | 381 | 5.0k | 6.7k | Informative articles | Long-form multiple-choice |
| Natural Questions (Kwiatkowski et al. (2019)) | 25k | 6.9k | 25k | Wikipedia articles | Span-based open-domain |
| NewsQA (Trischler et al. (2017)) | 658 | 8.5k | 120k | News articles (CNN) | Wh- and yes/no factual |

Table 1: Dataset statistics used in our evaluation. The datasets span diverse domains and query types, with NewsQA serving as an out-of-domain benchmark.

| Type | Method | QA Acc | Retrieval recall | Monetary cost($) | Latency (s) |
|------|--------|--------|-----------------|-----------------|-------------|
| Single | Fixed-size chunking (sentence) | 0.251 | 0.517 | 0.007 | 1.16 |
| | Fixed-size chunking (512) | 0.285 | 0.648 | 0.006 | 1.09 |
| | Late chunking (Günther et al. (2024)) | 0.363 | 0.661 | 0.007 | 1.26 |
| Multi | RAPTOR (Sarthi et al. (2024)) | 0.526 | 0.714 | 0.398 | 3.21 |
| | MAL RAG (Zheng et al. (2025)) | 0.561 | 0.842 | 0.301 | 4.14 |
| | GRAG (Edge et al. (2024)) | 0.547 | 0.806 | 0.269 | 4.20 |
| Ours | w/o $\mathcal{P}$ | 0.539 | 0.773 | 0.096 | 1.94 |
| | w/o $\mathcal{E}$(directly encode) | 0.427 | 0.723 | 0.079 | 2.00 |
| | w/o $\mathcal{E}$(summarize) | 0.582 | 0.861 | 0.204 | 3.85 |
| | SMARTCHUNK | 0.564 | 0.829 | 0.078 | 3.62 |

Table 2: Comparison of different chunking and retrieval strategies. Our method achieves competitive QA accuracy and retrieval recall while significantly reducing monetary cost and latency.

- **Retrieval Quality.** We report Recall@K, a widely used metric in RAG tasks, which measures whether the top-$K$ retrieved chunks contain the gold answer span. We set $K = 5$ in our experiments. This assesses the planner and retriever's ability to select relevant content under various chunking strategies.
- **Answer Quality.** To assess downstream QA performance, we adopt the LLM-as-a-judge approach for more nuanced evaluation, where a strong GPT-4o model judges whether the predicted answer is semantically correct with respect to the reference answer. This enables unified evaluation across datasets, accommodating variations in phrasing and the presence of multiple valid answers. Additionally, we report standard dataset-specific metrics (e.g., F1, ROUGE) in Appendix Table 9 for completeness.
- **System Efficiency.** To evaluate efficiency, we report two metrics: monetary cost and latency. Token-based prices are computed using provider rates (e.g., OpenAI, Together.AI Together AI (2024); OpenAI (2024)); details are in Section E. Notably, the cost of generator/summarizer models (e.g., GPT-4o) dominates, while small models like the planner and embedder are far cheaper and can even run locally. For consistency, we still include their cost using Together.AI pricing, offering a conservative estimate. While some applications prioritize monetary efficiency, latency is equally critical in real-time settings. SMARTCHUNK delivers strong performance across both cost and latency, adapting well to diverse deployment constraints.

**Baselines** We also compare against standard and state-of-the-art RAG systems.
- **Single-Level Chunking Baselines.** These methods use fixed chunk sizes without dynamic planning: Fixed-size chunking (sentence) (Kamradt (2024); LangChain team (2023); Liu & contributors (2022)) splits documents at sentence boundaries. Fixed-size chunking (512) (Lewis et al. (2020)) splits documents into non-overlapping 512-token chunks. Late chunking (Günther et al. (2024)) defers chunking until embeddings are computed, enabling better context preservation but still operates at a single granularity level.
- **Multi-Level Chunking without Planner or Compression.** These methods build multi-scale document hierarchies but lack adaptive planning or query-aware compression: MAL RAG (Zheng et al. (2025)) leverages chunks at multiple abstraction levels, but lacks both a planner and a Chunk Compression Encoder. As a result, it constructs the full chunk hierarchy for every document and relies on expensive GPT calls to generate summaries for each chunk. GRAG (Edge et al. (2024)) integrates graph-based representations for multi-level retrieval and reasoning. RAPTOR (Sarthi et al. (2024)) builds a multi-level chunk tree by recursively embedding, clustering, and summarizing text from the bottom up, and performs retrieval over this hierarchical structure.

Additional experimental details are shown in Section E.

## 5.2 MAIN RESULTS

The main results are presented in Table 2, with per-dataset breakdowns provided in Table 9. SMARTCHUNK consistently outperforms all baselines across key metrics. While single-level chunking baselines are low cost, their QA accuracy is substantially lower. SMARTCHUNK achieves up to about 30% improvement in accuracy over these methods. Compared to recent state-of-the-art tree- or graph-structured RAG methods that lack our proposed planner and chunk compression encoder, SMARTCHUNK delivers comparable or better results, showing an average gain of 1.7% in QA accuracy and 4.0% in retrieval recall. Most importantly, it does so with far greater efficiency, requiring

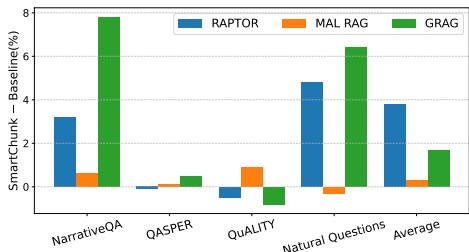
(a) QA quality among datasets.

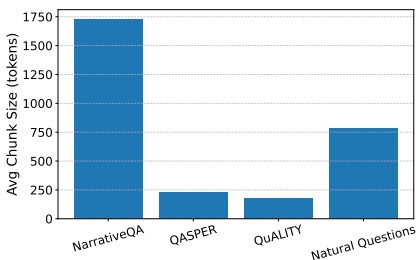
(b) Chunk sizes among datasets.

Figure 4: (a) Performance gaps of SmartChunk over competing methods on four benchmarks—NarrativeQA (ROUGE), QASPER (F1), QuALITY (Accuracy), and Natural Questions (F1); positive bars mean SmartChunk outperforms the baseline. (b) Average chunk sizes (tokens) selected by our planner across datasets, illustrating dataset-/query-adaptive behavior.

less than 30% of the GPT cost and achieving lower latency. This efficiency comes from our planner, which reduces the number of chunks that need to be built, and our compression module, which avoids repeated calls to GPT for summarization when constructing the chunk hierarchy.

To further assess the contributions of each module, we conduct ablation studies. Removing the planner leads to always building the full chunk tree, while removing the compression encoder requires GPT summarization for each chunk. Both of them result in higher cost and latency. Nevertheless, even these ablated versions of SMARTCHUNK remain competitive with existing baselines, confirming the effectiveness of our design. We also evaluate a variant that removes the compression encoder and directly encodes chunks without summarization. This leads to a noticeable drop in retrieval and QA performance, highlighting the importance of providing condensed representations to the retriever. These findings emphasize the value of our lightweight compression encoder, which produces effective summaries with far lower cost than querying large LMs.

**Total Cost of SMARTCHUNK** While SMARTCHUNK introduces additional training cost, we emphasize that this cost is incurred only once. The learned planner and compression encoder are universal components that can be directly applied across different datasets and domains, and we further demonstrate strong generalization to out-of-distribution data. In realistic RAG deployments, where the system serves millions of user queries, the one-time training cost is minor compared to the continuously accumulating inference cost of baseline methods(more than 5 times compare to ours). Also, we only fine-tune a lightweight 1.5B-parameter model (Qwen2.5-1.5B-Instruct), while the generator and summarizer in the RAG pipeline is a much larger GPT-class model whose inference

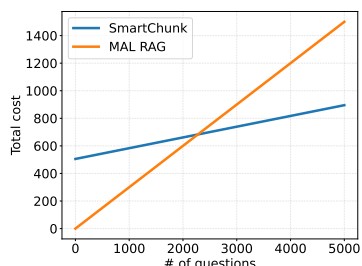
Figure 3: Total cost including training and test-time.

dominates the overall cost. In Figure 3, we visualize the total end-to-end cost of SmartChunk and MAL RAG as the number of questions increases. The total cost including both the training(data collection + training cost of compressor and planner) and test-time cost. While SmartChunk incurs a small fixed cost upfront due to planner and encoder training, its test-time cost grows very slowly. In contrast, MAL RAG begins at zero but scales fasr with the number of questions, surpassing SmartChunk at around 2000 queries. This illustrates that the training cost is negligible relative to the savings obtained through more efficient, query-adaptive chunking. Consequently, once amortized over a realistic query volume, SmartChunk becomes substantially more cost-effective.

### 5.3 ADDITIONAL OBSERVATIONS

**Robustness Across Document Types.** Our SMARTCHUNK framework demonstrates strong generalization across diverse document types, as shown in Figure 4a, which visualizes performance gaps relative to state-of-the-art baselines (positive values indicate SMARTCHUNK superiority). Specifically, the "performance gaps" refer to the accuracy improvement of SmartChunk over each state-of-the-art (SOTA) baseline, $\Delta_{gap} = \text{Performance}_{\text{SmartChunk}} - \text{Performance}_{\text{Baseline}}$. Consistent with results in the main table, SMARTCHUNK outperforms all baselines on average. GRAG performs well on knowledge-centric datasets like QuALITY but underperforms on narrative-driven tasks such as NarrativeQA, where SMARTCHUNK achieves nearly 8% higher accuracy. Similarly, RAPTOR struggles

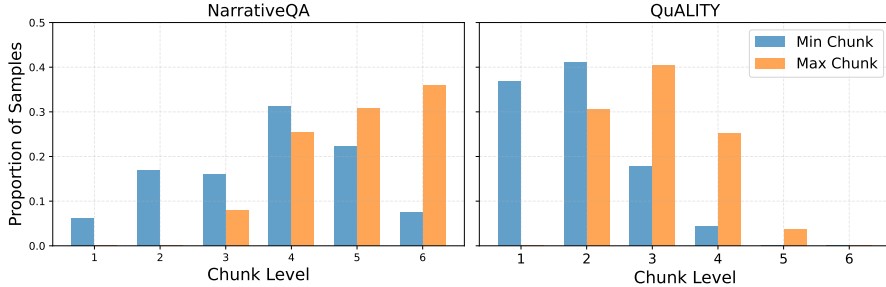

Figure 5: Distribution of Min–Max Chunk Sizes Across Datasets. Higher chunk levels correspond to larger chunk sizes (ranging from sentence-level to document-level).

| Method | F1 | Retrieval recall | Monetary cost($) | Latency (s) |
|---|---|---|---|---|
| Fixed-size chunking | 0.846 | 0.895 | 0.006 | 0.98 |
| MAL RAG (Zheng et al. (2025)) | 0.907 | 0.971 | 0.147 | 1.32 |
| SMARTCHUNK | 0.875 | 0.931 | 0.026 | 1.29 |
| SMARTCHUNK + few shot | 0.906 | 0.967 | 0.032 | 1.34 |

Table 3: Performance on out-of-domain dataset NewsQA.

with temporal reasoning, leading to a 3% gap on story-centric benchmarks NarrativeQA. MAL RAG shows stronger generalization to document structure changes but consistently incurs high cost due to exhaustive tree construction and summarization.

**Query and Dataset-Adaptive Chunking Behavior.** We analyze the chunk size decisions made by the planner across different datasets. As shown in Figure 4b, STITCH dynamically adjusts chunk sizes based on both document and query type. For instance, in NarrativeQA, which features long-form fictional stories paired with abstractive narrative comprehension questions, the planner selects larger chunks (avg. size: 1725) to capture broader contextual arcs necessary for narrative understanding. In contrast, on scientific datasets like QASPER, which require factual multi-hop reasoning over dense, technical papers, STITCH favors smaller chunks (avg. size: 230) to maintain precision and avoid context dilution. As shown in Figure 5, this adaptive behavior also manifests in the planner's predicted range of chunk levels. In NarrativeQA, both the minimum and maximum chunk levels are distributed toward the higher end (levels 4–6), reflecting the need for broader contextual integration to capture long-range narrative dependencies. In contrast, QuALITY exhibits a skew toward smaller chunk levels (1–3), as its questions typically target specific facts or short reasoning chains within individual paragraphs. The divergence between minimum and maximum chunk levels further suggests that STITCH learns to maintain flexible granularity. Starting from smaller, fine-grained chunks for grounding, then expanding to larger contextual spans when broader evidence aggregation is beneficial. This distributional pattern underscores the planner's ability to tailor chunk selection to dataset characteristics and query complexity, rather than relying on a fixed retrieval granularity.

**Out-of-Domain Generalization.** We evaluate the generalization capability of SMARTCHUNK on the out-of-distribution NewsQA (Trischler et al. (2017)). Results are shown in Table 3. Even without any in-domain fine-tuning, the planner trained on other datasets significantly outperforms the fixed-size chunking baseline in both QA F1 score (+2.8%) and retrieval recall (+3.7%), while maintaining a low monetary cost and latency. This indicates that the planner effectively transfers to new domains with different document structures and query distributions. Furthermore, when using few-shot prompting with only three in-context examples, SMARTCHUNK matches the performance of MAL RAG, the best-performing baselines in main experiments, while using only 25% of the monetary cost and comparable latency. This result highlights the robustness and adaptability of our proposed planner and compression encoder, even under domain shift, and demonstrates the practicality of our method in real-world applications where training data may be limited or distribution shifts are common.

**On the Necessity of Reasoning and Finetuning for Planner** As shown in Table 4, frozen models perform poorly, with low planning accuracy (42.6% even with few-shot context) and high latency due to the lack of training for length control. Finetuned classifiers also remain only moderately accurate. In contrast, our fine-tuned planner achieves the best performance, improving accuracy by 9.4% over the strongest baseline while maintaining reasonable latency. These results demonstrate the necessity of combining finetuning and reasoning: pretrained LLMs fail to adapt without task-

| Type | Method | Planning Accuracy | Planning Latency(s) |
|---|---|---|---|
| no Finetuning | LLM | 0.407 | 2.472 |
| | LLM + Few shot | 0.426 | 4.036 |
| | SLM | 0.382 | 3.058 |
| no Reasoning | MLP | 0.609 | 0.0003 |
| | SLM | 0.724 | 0.019 |
| Finetuning + Reasoning | Ours | 0.820 | 0.848 |

Table 4: Ablation study showing the effect of planner, finetuning, and reasoning components on planning accuracy, cost, QA accuracy, and latency. Our full system achieves the best performance across all metrics.

| Method | # Training Tokens | Planning Acc | Planning Latency | QA Acc | Monetary cost($) |
|---|---|---|---|---|---|
| SFT | 795k | 0.740 | 1.162 | 0.491 | 0.085 |
| RL | / | 0.356 | 0.365 | 0.427 | 0.066 |
| SFT+RL | 795k | 0.763 | 0.831 | 0.538 | 0.073 |
| SFT+RL | 418k | 0.544 | 0.693 | 0.467 | 0.081 |
| STITCH | 418k | 0.820 | 0.848 | 0.564 | 0.078 |

Table 5: Comparison of training strategies for the planner. The details of baselines are shown in Section E

specific training, while classifiers without reasoning cannot capture the structured decision-making required. Our planner instead learns to reason over metadata and query intent, producing latency-aware plans tailored to the task.

**Effectiveness and Efficiency of STITCH.** We evaluate the effectiveness and efficiency of STITCH in Table 5. STITCH achieves nearly 5% higher accuracy than the strongest SFT+RL baseline while using only half the supervised tokens. This is an important advantage given the high cost of labeled data. It also reduces monetary cost, as RL can explore beyond suboptimal pseudo-labels. Standard RL, however, often fails due to weak base models and instability under multi-objective rewards. In contrast, STITCH balances accuracy, latency, and cost effectively, with gains driven by solving first with RL to cut reliance on supervision, and targeted imitation learning to stabilize training where RL struggles. We further analyze the training dynamics of STITCH in Appendix Figure 7 to demonstrate the STITCH's stability. Beyond these gains, we also demonstrate STITCH's ability to generalize to new domains such as mathematical reasoning shown in Section F.

**Orthogonality to Other RAG Advances** We demonstrate that SMARTCHUNK is orthogonal to recent advances in RAG pipelines, such as late chunking (Günther et al. (2024)) and hybrid search (Liu & contributors (2022)). Hybrid search combines both keyword-based methods (BM25 Roberts et al. (2020)) and vector (embedding) search techniques. As shown in Figure 6, integrating SMARTCHUNK with these methods gets further gains in QA accuracy. This indicates that SMARTCHUNK can complement complementary improvements in chunking and retrieval, amplifying their effectiveness when combined.

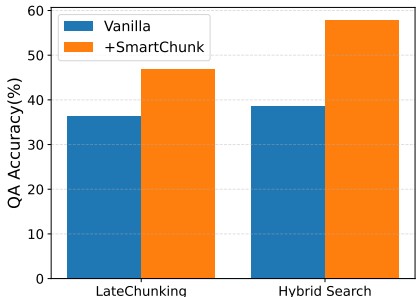

Figure 6: QA accuracy when combining SMARTCHUNK with other RAG improvements.

## 6 CONCLUSIONS

We introduced SmartChunk Retrieval, a query-adaptive framework for long-document QA that dynamically selects chunk granularity and compresses embeddings to balance accuracy and efficiency. Across diverse benchmarks, SmartChunk consistently outperforms static baselines while reducing retrieval noise and cost. Our results highlight the importance of flexible retrieval strategies that adapt to both document structure and query demands. Beyond QA, this framework points toward a general direction for controllable, resource-aware reasoning in language models.

**Future work** In future work, we plan to explore SmartChunk's application to deep research and other open-book QA tasks. Beyond QA, STITCH also points toward a broader direction for controllable, resource-aware reasoning in language models. Another future research direction we would like to explore is image-text understanding using STITCH in the shared embedding space to perform multimodal document retrieval.

ACKNOWLEDGEMENTS

This work was conducted during an internship at Adobe. It is also supported in part by the National Science Foundation grants CCF-2550179, CCF-2403075, CCF-2212426, the Office of Naval Research grant N000142412289.

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

## A    APPENDIX

**The Supplementary Material is organized as follows:**

- In Section B, we describe the role of LLMs in this paper.
- In Section C, we provide ethics statement.
- In Section D, we provide Reproducibility statement.
- In Section E, we explain the additional experimental details.
- In Section F, we conduct additional ablation experiments show the effective of each module.
- In Section G, we show the pseudo code of STITCH.
- In Section H, we discuss why we need different chunk size and indicate that static chunking–retrieval design is the primary bottleneck as retrieval quality is highly sensitive to chunk size, and no single granularity works across heterogeneous queries and document structure.
- In Section I, we discuss the design choices and justifications.
- In Section J, we discuss the limitation.

## B    THE USE OF LLMS

In this paper, we leverage large language models (LLMs) to assist with writing and refinement. Specifically, LLMs are used to modify, restructure, and improve the clarity and fluency of our text.

## C    ETHICS STATEMENT

Our study does not involve human subjects, personally identifiable information, or sensitive user data. All datasets used are publicly available and widely adopted in the research community. As for the potential risks of misuse, since our method focuses on improving the efficiency and accuracy of retrieval-augmented QA systems, we believe it will not lead to direct harmful applications.

## D    REPRODUCIBILITY STATEMENT

The full details of model architecture, training setup, hyperparameters, and evaluation protocols are provided in the main text and appendix. All datasets used are publicly accessible, and we describe the preprocessing pipeline in Section E.

## E    EXPERIMENTAL DETAILS

**Models** We use GPT-4o (Hurst et al. (2024)) as the end answer generation and summarization model across all method. The planner is fine-tuned from the Qwen2.5-1.5B-Instruct (Yang et al. (2025)) model. For retrieval, we adopt a standard vector-based retrieval setup. We use SBERT (multi-qa-mpnet-base-cos-v1) (Reimers & Gurevych (2019)) as the baseline embedding model. Our Chunk Compression Encoder is initialized from the same SBERT model and fine-tuned to produce compressed representations for high-level chunks.

**Hardware** For the hardware requirements, all of experiments are done with 8 80G H100 GPUs.

**Training settings.** We adopt the verl framework (Sheng et al. (2024)) for training the planner. We utilize the Adam optimizer (Kingma (2014)) with a constant learning rate of $1 \times 10^{-6}$. For rollout, the prompt batch size is 256 and we sample 8 responses for each prompt. For training, the minibatch size is 64. To find the appropriate hint length in the hinted RL rollout step, we split the expert trace into sentences.

**Monetary cost.** We report the API price we used to compute the monetary code for each model.

| Platform | Model | Input | Output |
|----------|-------|-------|--------|
| OpenAI | GPT-4o | 2.5 | 10 |
| OpenAI | Embedding | 2.5 | 10 |
| Together.AI | Qwen2.5-7B | 0.3 | 0.3 |
| Together.AI | Llama-3.1 8b | 0.18 | 0.18 |
| Together.AI | GPT-oss-120b | 0.15 | 0.15 |
| Together.AI | GPT-oss-20b | 0.05 | 0.05 |

Table 6: The monetary cost of each model.

**Dataet train/test split.** If the dataset is already partitioned into non-overlapping training, validation, and test portions, we will use it. Or, the dataset is split into 80% for training and 20% for evaluation. The training portion is used to train both the planner and the chunk compression encoder modules in our system.

**Planner training baseline** We provide the baseline details in Table 5. For RL, we adopt DAPO (Yu et al. (2025)), an improved GRPO variant. SFT is trained for one epoch, following standard practice in prior work (Muennighoff et al. (2025)), while RL is trained for 120 steps with the same configuration as STITCH for fair comparison. During rollout, we use a prompt batch size of 256 and sample 8 responses per prompt; for training, the mini-batch size is set to 64.

**Prompt** We also list the prompt we use below to provide better reproducibility.

- **Summarizer:** *"Summarize the following text in at most $n$ tokens: [paragraph]."* Here, $n$ is a hyperparameter we can get. We use 128 in our experiments as this is the commonly used working length of the embedding model. [paragraph] is the content we want to summarize.

- **Planner:** *"You are an expert in analyzing document structure for retrieval.*
  *Input: - A document (up to 1000 tokens) - A question about the document*
  *Task: Predict two values: 1. The \*\*largest chunk size\*\* that still provides sufficient context to answer the question. 2. The \*\*smallest chunk size\*\* that still preserves the necessary information for the answer.*
  *Valid chunk sizes: ["document", "section", "subsection", "paragraph", "sentence"].*
  *Output: Please analyze the context, then return your prediction in JSON format with the fields: { "largest_chunk_size": "...", "smallest_chunk_size": "..." }*
  *Context: \*\*Document (first 1000 tokens):\*\* [document text] \*\*Question:\*\* [question text]"*

## F ABLATION RESULTS

**Planner input and output** Here is an example in QASPER (Dasigi et al. (2021)).

- **Input** *Input: - First 1000 tokens of the document - A question about the document Task: Predict two values: 1. The \*\*largest chunk size\*\* that still provides sufficient context to answer the question. 2. The \*\*smallest chunk size\*\* that still preserves the necessary information for the answer. Valid chunk sizes: ["document", "section", "subsection", "paragraph", "sentence"]. Output: Please analyze the context, then return your prediction in JSON format with the fields: { "largest_chunk_size": "...", "smallest_chunk_size": "..." } Context: \*\*Document (first 1000 tokens):\*\* [It has been shown that word embeddings derived from large corpora tend to incorporate biases present in their training data. Various methods for mitigating these biases have been proposed, but recent work has demonstrated that these methods hide but fail to truly remove the biases, which can still be observed in word nearest-neighbor statistics. In this work we propose a probabilistic view of word embedding bias. We leverage this framework to present a novel method for mitigating bias which relies on probabilistic observations to yield a more robust bias mitigation algorithm. We demonstrate that this method effectively reduces bias according to three separate measures of bias while maintaining embedding quality across various popular benchmark semantic tasks] \*\*Question:\*\* [How is embedding quality assessed?]*

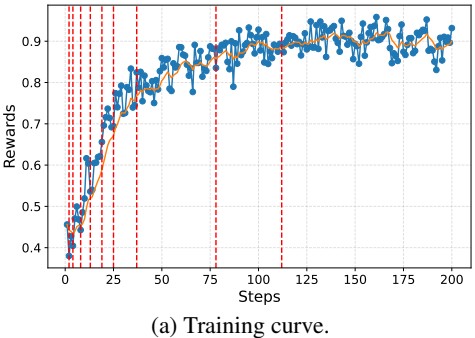 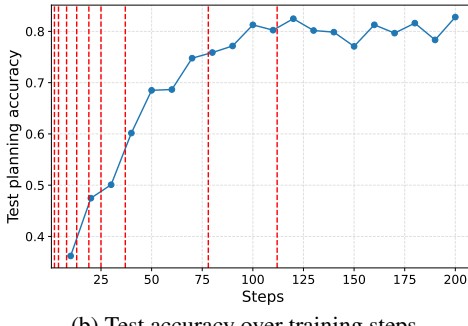

(a) Training curve.         (b) Test accuracy over training steps

Figure 7: Training dynamics of STITCH. The blue curves represent the raw training reward (left) and test planning accuracy (right) over training steps. In the left figure, the orange curve shows a time-weighted EMA of the training reward, highlighting the overall trend. Red dashed lines show the points where targeted SFT is applied.

- **Output** *Question seeks how embedding quality is measured. Document states quality is maintained "across various popular benchmark semantic tasks." This key detail appears in one sentence, so smallest chunk is sentence. However, understanding the context of bias mitigation requires the surrounding discussion, making section the largest relevant unit. Retrieval at document level is too broad; section provides enough scope. {*  *"largest_chunk_size": "section", "smallest_chunk_size": "sentence" }*

**Training curve** We further analyze the training dynamics of STITCH in Figure 7. The results demonstrate that STITCH is both stable and effective. Training reward increases steadily and converges to a high level. Test accuracy follows a similar trajectory, showing consistent improvement over training and eventually plateauing near $80\%$. Importantly, the alternating use of RL and SFT ensures robust progress: RL explores beyond pseudo-labels, while SFT injections provide stability at critical points, preventing collapse. Overall, the curves confirm that STITCH achieves strong and stable learning dynamics.

**Trace diversity** We also demonstrate the importance of using diverse reasoning trace for finetuning planner.

| Method | # of samples | # of tokens | Planning Acc |
|---|---|---|---|
| Base | / | / | 41.7% |
| SFT (single model, small) | 2000 | 931k | 53.1% |
| SFT (single model, large) | 2000 | 783k | 45.6% |
| SFT (6 models) | 2000 | 794k | 74.3% |
| SFT (6 models) | 1000 | 401k | 69.2% |
| STITCH | / | **421k** | **81.8%** |

Table 7: Comparison of planner training methods. STITCH achieves the highest planning accuracy while using fewer tokens.

**Hint Extraction in STITCH.** In Step 2 of STITCH, we extract a hint from the expert-generated reasoning trace to guide hinted rollouts. We evaluate several strategies for determining the hint length and find that estimating it based on the average success rate among all samples achieves the best balance between performance and computational cost. The results are shown in Figure 8. This strategy provides strong guidance while minimizing overhead, and is therefore adopted across all experiments, including those in the main results.

**STITCH generalizes to other domains such as mathematical reasoning.** STITCH outperforms all the baselines, the resukts are presented in Table 8. Let us discuss each baseline in turn. STITCH improves final accuracy by almost 20% over vanilla `GRPO`. SFT can offer a stronger starting point for RL, but STITCH is still better.

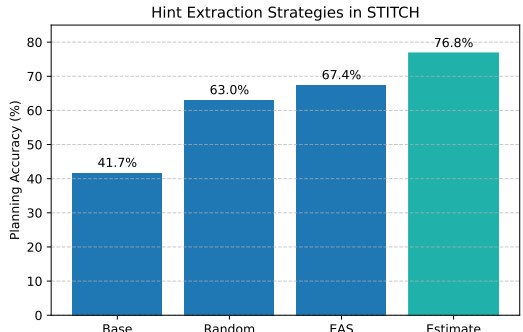

Figure 8: Comparison of hint extraction strategies used in Step 2 of the STITCH framework. Evaluation is conducted under a fixed training budget of 204,800 rollouts (8 rollouts, 200 steps, batch size 128 using vanilla GRPO). The `Estimate` strategy, which adaptively determines hint length based on rollout success rates, achieves the highest planning accuracy while maintaining low computational cost. This strategy is adopted for all remaining experiments.

| Model | Dataset | GRPO | SFT+GRPO | STITCH | DeepSeek-Distill |
|---|---|---|---|---|---|
| Qwen-1.5B-Instruct | NuminaMath-CoT | 0.347 | 0.348 | 0.366 | 0.368 |
| Qwen-1.5B-Instruct(IL buffer size T = 64) | NuminaMath-CoT | 0.347 | 0.348 | 0.367 | 0.368 |
| Qwen-3B-Instruct | NuminaMath-CoT | 0.475 | 0.503 | 0.649 | / |

Table 8: SMARTCHUNK outperforms other baselines and nearly reaches the accuracy of DeepSeek-R1-Distill, which has the benefit of vast training data. There is no DeepSeek-R1-Distill for Qwen-3B provided by (Guo et al. (2025)), so its cells are left blank.

**STITCH succeeds on the hard questions, where other baselines fail.** To understand the gains of STITCH, we train and evaluate each method exclusively on very difficult problems. The goal is to investigate whether expert hints in STITCH can help SLMs learn new information particularly from these hard questions. We conduct the experiment on the NuminaMath-CoT dataset and the Qwen2.5-3B-Instruct as the base model. To build the hard dataset, we first run ordinary SFT and from the training dataset, we select the 500 questions for which three independent generations produce no correct trace (pass@3 = 0). These tasks are unsolved by the small model, and their expert traces proved too complex for SFT to learn. The 500 samples were split into 80/20 train/test subsets. We then again used Qwen2.5-3B-Instruct as the base model and trained each method on this hard subset. The results are shown in Figure 9a. All baselines show little or no improvement on the test set after training on the hard questions. In contrast, STITCH achieves a clear upward performance with continued training, demonstrating that its partial expert guidance and branched rollout strategy can provide learnable information, even when standard `SFT` and vanilla `GRPO` fail. We argue that STITCH succeeds because it adaptively reduce problem difficulty and densifies the rewards. As shown in Figure 9b, STITCH sharply lowers the solve-none ratio, getting more informative samples and richer feedback. This enables STITCH to learn effectively even from very hard questions and complex reasoning traces.

**Scalability of SmartChunk** While our current evaluation focuses on moderate-sized corpora for controlled comparison with prior RAG baselines, SmartChunk is inherently scalable because the planner operates at the chunk-level index rather than the token level. This design allows efficient retrieval decisions even when the underlying corpus grows by orders of magnitude.

| Type | Method | NarrativeQA(ROUGE) | QASPER(F1) | QuALITY(Acc) | Natural Questions(F1) | Avg (Accuracy) |
|---|---|---|---|---|---|---|
| Single | Fixed-size chunking (512) | 0.409 | 0.496 | 0.695 | 0.727 | 0.285 |
| | Late chunking (Günther et al. (2024)) | 0.421 | 0.503 | 0.738 | 0.756 | 0.363 |
| Multi | RAPTOR (Sarthi et al. (2024)) | 0.442 | 0.584 | 0.824 | 0.758 | 0.526 |
| | MAL RAG (Zheng et al. (2025)) | 0.468 | 0.582 | 0.810 | 0.809 | 0.561 |
| | GRAG (Edge et al. (2024)) | 0.396 | 0.578 | 0.827 | 0.742 | 0.547 |
| Ours | SMARTCHUNK | 0.474 | 0.583 | 0.819 | 0.806 | 0.564 |

Table 9: QA performance for each dataset separately.

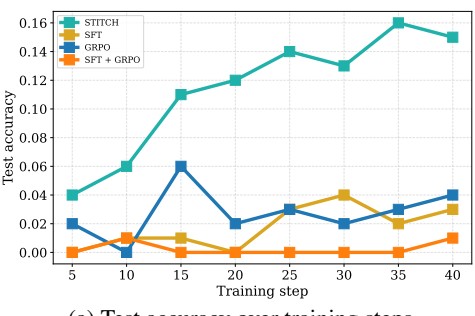 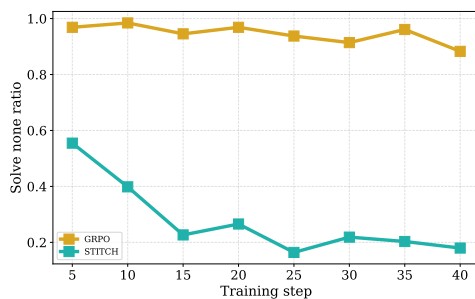

(a) Test accuracy over training steps    (b) Ratio of questions whose rollouts are all wrong

Figure 9: (a) Test accuracy on very hard questions over training steps. SMARTCHUNK outperforms other baselines significantly while the traditional methods (SFT, GRPO, SFT+GRPO) learn little. (b) Proportion of test questions for which every rollout fails (solve-none ratio). In vanilla GRPO, the ratio stays persistently high, signalling that training stalls under sparse rewards. In SMARTCHUNK, the solve-none ratio for regular rollout starts similarly high, but hinted RL injects expert traces and densifies the reward. This rate continuously decreases, confirming that the model is learning.

---

**Algorithm 1** SMARTCHUNK: SmartChunk Test-Time Inference Pipeline

---

**Require:** Query $q \in \mathcal{Q}$, reference corpus $D$, base chunk set $C$, planner $\mathcal{P}$, Chunk Compression Encoder $\mathcal{E}$, retriever $\mathcal{R}$, generator $\mathcal{G}$

**Ensure:** Answer $\hat{a}$

1: **// Step 1: Plan chunk-size range**
2: $(s_{\min}, s_{\max}) \leftarrow \mathcal{P}(q, \text{metadata}(D))$ ▷ predict smallest and largest chunk sizes to consider
3: **// Step 2: Build multi-level hierarchy (in parallel with planning)**
4: $\mathcal{H} \leftarrow \mathcal{E}(C, s_{\min}, s_{\max})$ ▷ $\mathcal{H}$ is a hierarchy of chunks with multi-level embeddings
5:     ▷ smaller chunks are clustered and aggregated into higher-level chunks
6:     ▷ each high-level chunk embedding encodes the semantics of its cluster
7: **// Step 3: Retrieve relevant chunks (vanilla RAG over multi-level embeddings)**
8: $S \leftarrow \mathcal{R}(q, \mathcal{H}, \text{top} = k)$ ▷ retrieve top-$k$ chunks using multi-level embeddings
9: **// Step 4: Generate answer (vanilla RAG)**
10: $\hat{a} \leftarrow \mathcal{G}(q, S)$
11: **return** $\hat{a}$

---

In practice, SmartChunk can be seamlessly integrated with sharded vector databases (e.g., FAISS Douze et al. (2024), ScaNN Sun et al. (2023b)) and supports parallel retrieval across chunk hierarchies, enabling real-time performance. The adaptive planner also reduces the number of retrieval calls per query, lowering cost and latency as corpus size increases.

# G   ALGO

## G.1   SMARTCHUNK

**Test-time pseudo-code**

## G.2   STITCH

**RL Reward Components in STITCH** To train the planner effectively, we design a multi-objective reward $R$ that balances several desiderata:

$$R = R_{\text{QA}} + R_{\text{Cost}} + R_{\text{Format}} + R_{\text{Length}}$$

Each term is defined as follows:

- QA Answer Correctness $R_{\text{QA}}$: $R_{\text{QA}} = \begin{cases} +1, & \text{if generated answer } a = a^{\text{gold}} \\ 0, & \text{otherwise} \end{cases}$

- Chunk Budget Penalty $R_{\text{Cost}}$: To encourage fewer, more informative chunks, we penalize the cost of building multi-level chunks $c$: $R_{\text{Cost}} = -\alpha \cdot \text{ChunkCost}$

- Trace Length Penalty $R_{\text{Length}}$: We penalize overly long traces beyond a target token budget $L_{\max}$. Let $\ell$ be the number of generated tokens in the reasoning trace: $R_{\text{Length}} = -\beta \cdot \text{ReLU}(\ell - L_{\max})$

- Format $R_{\text{Format}}$: To encourage structured reasoning and ensure the answer is extractable, we include a reward term that promotes outputs matching a predefined format. This is standard in reasoning-based models training. $R_{\text{Format}} = \begin{cases} +\gamma, & \text{if matches format} \\ 0, & \text{otherwise} \end{cases}$

- Pseudo-Label Alignment $R_{\text{PlanningAcc}}$: As an alternative to optimizing QA accuracy and efficiency tradeoff($R_{\text{QA}} + R_{\text{Cost}}$), we introduce a reward that aligns the planner's output with pseudo-labels. This bypasses retrieval and answer generation during training, enabling more stable and fast training. This term is particularly useful during early training to guide the model before it begins fully exploring the performance-efficiency trade-off on its own.

**Training objective** Our training objective follows the formulation of Shao et al. (2024), and use the same formulation. For each question $q$, STITCH samples a group of outputs $\{o_1, o_2, ...o_G\}$ from the old policy $\pi_{old}$, and then optimizes the policy model by maximizing the following objective:

$$\mathcal{J}_{\text{STITCH}}(\theta) = \mathbb{E}_{(q,\rho,a)\sim\mathcal{D}, \{o_i\}_{i=1}^{G}\sim\pi_{\text{old}}(\cdot|(q,\rho))}$$

$$\left[ \frac{1}{G}\sum_{i=1}^{G}\frac{1}{|o_i|}\sum_{t=1}^{|o_i|}\left(\min\left(r_{i,t}(\theta)\hat{A}_{i,t}, \text{clip}\left(r_{i,t}(\theta), 1-\varepsilon, 1+\varepsilon\right)\hat{A}_{i,t}\right) - \beta D_{\text{KL}}(\pi_\theta||\pi_{\text{ref}})\right) \right], \quad (3)$$

where

$$r_{i,t}(\theta) = \frac{\pi_\theta(o_{i,t}|q,\rho,o_{i<t})}{\pi_{\text{old}}(o_{i,t}|q,\rho,o_{i<t})}, \quad \hat{A}_{i,t} = \frac{R_i - \text{mean}(\{R_i\}_{i=1}^{G})}{\text{std}(\{R_i\}_{i=1}^{G})} \quad (4)$$

where $\varepsilon$ and $\beta$ are hyper-parameters, and $\hat{A}_{i,t}$ is the advantage calculated based on relative rewards of the outputs inside each group only. $\pi_\rho$ and $\pi_{\text{old}}$ are the current and old policy models. $\varepsilon$ is a clipping-related hyper-parameter.

STITCH regularizes by directly adding the KL divergence between the trained policy and the reference policy to the loss. We estimate the KL divergence with the following unbiased estimator:

$$\mathbb{D}_{\text{KL}}[\pi_\theta \| \pi_{\text{ref}}] = \frac{\pi_{\text{ref}}(o_{i,t} \mid q, o_{i,<t})}{\pi_\theta(o_{i,t} \mid q, o_{i,<t})} - \log\frac{\pi_{\text{ref}}(o_{i,t} \mid q, o_{i,<t})}{\pi_\theta(o_{i,t} \mid q, o_{i,<t})} - 1. \quad (5)$$

The key difference from GRPO lies in the input and reward function: GRPO takes only the query as input, whereas our method uses the document along with related questions.

**STITCH pseudo code** We also show the pseudo code in Algorithm 2

---

**Algorithm 2** STITCH: Solve with RL, Then Imitate To Close Holes

---

**Require:** Dataset $\mathcal{D}$, planner policy $\mathcal{P}_\theta$, batch size $N$, GRPO optimizer, imitation learning (IL) buffer size $T$
1: Initialize $B_{\text{IL}} \leftarrow \emptyset$
2: **for** each training step **do**
3:     Sample batch $\{(q_j, D_j, a_j)\}_{j=1}^N \sim \mathcal{D}$
4:     Initialize RL buffer $B_{\text{RL}} \leftarrow \emptyset$
5:     **for** each $(q, D, a)$ in batch **do**
6:         Extract metadata $m \leftarrow \text{METADATA}(D)$
7:         # Step 1: Vanilla RL rollout #
8:         Sample vanilla rollouts $\{o_i\}_{i=1}^G \sim \mathcal{P}_\theta(\cdot \mid q, m)$
9:         **if** IsSOLVABLE$(\{o_i\}, a)$ **then**
10:             Add $(q, m, \{o_i\})$ to $B_{\text{RL}}$
11:             **continue**
12:         **end if**
13:         # Expert trace generation #
14:         Get pseudo-label $y \leftarrow \text{GETCHUNKLABEL}(q, D, a)$
15:         Generate reasoning trace $\tau \leftarrow \text{GENTRACE}(q, D, y)$
16:         # Step 2: Hinted RL rollout #
17:         Generate hint $\rho \leftarrow \text{EXTRACTHINTTRACE}(\tau)$
18:         Sample hinted rollouts $\{o_i'\}_{i=1}^G \sim \mathcal{P}_\theta(\cdot \mid q + \rho, m)$
19:         **if** IsSOLVABLE$(\{o_i'\}, a)$ **then**
20:             Add $(q + \rho, m, \{o_i'\})$ to $B_{\text{RL}}$
21:         **else**
22:             Add $(q, D, a, \tau)$ to $B_{\text{IL}}$
23:         **end if**
24:     **end for**
25:     # Policy update via GRPO #
26:     **if** $B_{\text{RL}} \neq \emptyset$ **then**
27:         UPDATEPOLICYGRPO$(\pi_\theta, B_{\text{RL}})$
28:     **end if**
29:     # Step 3: Imitation learning on unsolved cases #
30:     **if** $|B_{\text{IL}}| \geq T$ **then**
31:         IMITATIONLEARNINGSTEP$(B_{\text{IL}})$
32:         Clear $B_{\text{IL}}$
33:     **end if**
34: **end for**

---

## H   MULTI-LEVEL CHUNK HIERARCHY

We provide an example of the Multi-level chunk hierarchy. The comparison between multi-sentence–level and paragraph-level summarization highlights the trade-offs in chunk granularity. Multi-sentence chunks preserve finer detail, capturing multiple aspects of the original text. This granularity leads to richer reasoning traces and ensures that subtle arguments—like the distinction between geometric and probabilistic treatments—are not lost. However, it also produces longer outputs and potentially introduces redundancy across chunks.

In contrast, paragraph-level summarization yields more concise results by collapsing related content into a single compact representation. While this improves efficiency and reduces redundancy, some nuance is sacrificed: details on experimental setup and fairness principles are compressed, making the summary less expressive about specific mechanisms.

Taken together, the results illustrate why static chunking at a single level is suboptimal. Finer chunks capture more information but may be inefficient, whereas coarser chunks are efficient but risk oversimplification. SmartChunk's query-adaptive planning balances these extremes, allocating finer granularity when detailed reasoning is necessary and coarser granularity when efficiency suffices, leading to both accurate and cost-effective retrieval.

**Whole paragraph** *paragraphs": [ [ "Word embeddings, or vector representations of words, are an important component of Natural Language Processing (NLP) models and necessary for many downstream tasks. However, word embeddings, including embeddings commonly deployed for public use, have been shown to exhibit unwanted societal stereotypes and biases, raising concerns about disparate impact on axes of gender, race, ethnicity, and religion BIBREF0, BIBREF1. The impact of this bias has manifested in a range of downstream*

*tasks, ranging from autocomplete suggestions BIBREF2 to advertisement delivery BIBREF3, increasing the likelihood of amplifying harmful biases through the use of these models.", "The most well-established method thus far for mitigating bias relies on projecting target words onto a bias subspace (such as a gender subspace) and subtracting out the difference between the resulting distances BIBREF0. On the other hand, the most popular metric for measuring bias is the WEAT statistic BIBREF1, which compares the cosine similarities between groups of words. However, WEAT has been recently shown to overestimate bias as a result of implicitly relying on similar frequencies for the target words BIBREF4, and BIBREF5 demonstrated that evidence of bias can still be recovered after geometric bias mitigation by examining the neighborhood of a target word among socially-biased words.", "In response to this, we propose an alternative framework for bias mitigation in word embeddings that approaches this problem from a probabilistic perspective. The motivation for this approach is two-fold. First, most popular word embedding algorithms are probabilistic at their core – i.e., they are trained (explicitly or implicitly BIBREF6) to minimize some form of word co-occurrence probabilities. Thus, we argue that a framework for measuring and treating bias in these embeddings should take into account, in addition to their geometric aspect, their probabilistic nature too. On the other hand, the issue of bias has also been approached (albeit in different contexts) in the fairness literature, where various intuitive notions of equity such as equalized odds have been formalized through probabilistic criteria. By considering analogous criteria for the word embedding setting, we seek to draw connections between these two bodies of work.", "We present experiments on various bias mitigation benchmarks and show that our framework is comparable to state-of-the-art alternatives according to measures of geometric bias mitigation and that it performs far better according to measures of neighborhood bias. For fair comparison, we focus on mitigating a binary gender bias in pre-trained word embeddings using SGNS (skip-gram with negative-sampling), though we note that this framework and methods could be extended to other types of bias and word embedding algorithms." ],*

**Multi-sentence level summarization**  *"Word embeddings are vital for NLP tasks but have been shown to encode stereotypes and biases across gender, race, ethnicity, and religion BIBREF0, BIBREF1. Such biases can propagate into downstream applications, including autocomplete BIBREF2 and ad delivery BIBREF3, amplifying harmful social inequalities and raising concerns about fairness and disparate impact in real-world systems."*

*"Bias in word embeddings is often mitigated by projecting words onto a bias subspace (e.g., gender) and removing residual components BIBREF0. Bias is typically measured with the WEAT statistic BIBREF1, which compares cosine similarities between groups. However, WEAT may overestimate bias due to frequency effects BIBREF4, and studies show that even after geometric debiasing, bias remains detectable in neighborhoods of socially-biased words BIBREF5."*

*"We propose a probabilistic framework for mitigating bias in word embeddings, motivated by two factors. First, embeddings are inherently probabilistic, trained to model word co-occurrence probabilities BIBREF6, so bias treatment should reflect both geometric and probabilistic aspects. Second, fairness literature often formalizes equity using probabilistic criteria, such as equalized odds. By adapting analogous principles, our approach connects embedding bias mitigation with established fairness concepts, offering a unified and principled perspective."*

*"Experiments on bias mitigation benchmarks show our framework matches state-of-the-art methods in geometric debiasing while significantly outperforming them in reducing neighborhood bias. For evaluation, we target binary gender bias in pre-trained SGNS (skip-gram with negative sampling) embeddings. Importantly, the framework is general and can be extended to address other forms of bias and applied across different word embedding algorithms."*

**Paragraph level summarization** *Word embeddings are essential for NLP but encode societal biases that propagate into applications like autocomplete and ad delivery, raising fairness concerns BIBREF0–BIBREF3. Traditional debiasing projects words onto a bias subspace and measures with WEAT BIBREF0, BIBREF1, though WEAT may overestimate bias and residual signals persist BIBREF4, BIBREF5. We introduce a probabilistic framework that accounts for embeddings' probabilistic nature and adapts fairness notions like equalized odds BIBREF6. Experiments show our method performs comparably in geometric debiasing and substantially better in mitigating neighborhood bias, evaluated on binary gender bias in SGNS but extensible to other biases and embedding algorithms.*

This also help us explain why we need different chunk size. In short, contiguous spans rarely suffice for long-range reasoning. For example, a typical question in NarrativeQA (Kočiskỳ et al. (2018)) is "What motivates the main character's change of heart by the end?". This is not answerable by extracting localized text. It requires integrating information spread widely across the entire text. While oversized chunks introduce a different problem: important fine-grained details are blurred or lost, making it harder to ground accurate answers. At the same time, embedding models have limited effective context and degrade on very long inputs, while LLMs remain inefficient at leveraging long context. Not all queries require long context in the first place. For example, in the QASPER dataset of questions about NLP research papers (Dasigi et al. (2021)), many answers are extractive and confined to a single paragraph. A question such as "What other political events are included in the database?" can typically be answered from a sentence- or paragraph-level chunk in the methods section,

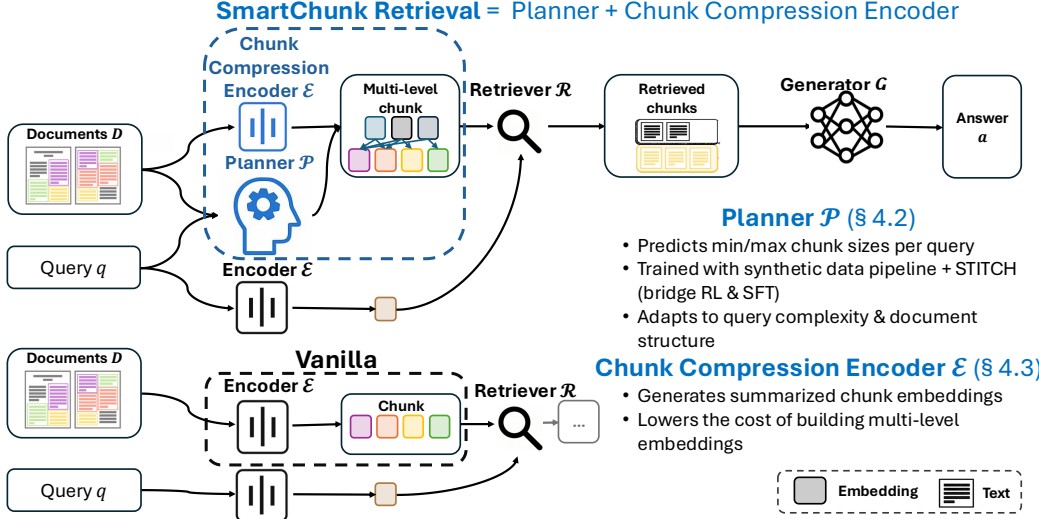

Figure 10: Overview of the SMARTCHUNK framework. Compared to vanilla RAG, which uses fixed chunking and flat retrieval, SmartChunk introduces two key modules: (1) a planner $\mathcal{P}$ that predicts the smallest and largest chunk sizes per query, enabling adaptive multi-level retrieval, and (2) a Chunk Compression Encoder $\mathcal{E}$ that produces compact, high-level embeddings for aggregated spans, lowering the cost of multi-level representation. These additions allow SMARTCHUNK to adapt chunking to query complexity and document structure, balancing accuracy and efficiency. Modules added by SmartChunk are shown in blue, while baseline modules from vanilla RAG are shown in black. The figure distinguishes between text (represented as blocks with horizontal lines) and embeddings (shown as solid-colored blocks).

without needing the entire paper as context. Using overly large chunks for such questions can retrieve the relevant information with surrounding noise, hurting retrieval accuracy and answer quality.

## I  DESIGN CHOICES AND JUSTIFICATIONS.

**Planner**

*(1) Planning as More Than Classification.* At first glance, predicting planning signals such as smallest or largest chunk sizes may seem like a straightforward classification problem. But simple classification ignores the structured and sequential nature of planning (London et al. (2016)), which involves balancing a trade-off between retrieval cost and answer quality. Unlike classification, a reasoning model can capture this ambiguity through explicit traces that encode conditional logic and deep thinking (e.g., factual queries favor finer chunks, while synthesis tasks require larger spans). Theoretical work in structured prediction demonstrates that introducing intermediate latent variables reduces sample complexity and improves generalization under non-linear mappings (Bello & Honorio (2018); Li & Liu (2021)), suggesting that reasoning traces serve as scaffolding to stabilize learning and improve robustness to distribution shift. Our ablation experiments further confirm this: simple MLP routers or LLMs without reasoning underperform compared to a finetuned SLM, showing the necessity of explicit reasoning for planning. Most importantly, reasoning-based planners exhibit stronger transfer across tasks and domains, consistent with insights from compositional generalization (Wei et al. (2022); Sanford et al. (2024)).

*(2) Add hint to build curriculum learning* Reasoning problems are inherently compositional hence we need a chain-of-thought (Wei et al. (2022)) to solve such problems. Suppose the problem contains $T$ steps/subtasks to solve, each with successful completion probability of $\epsilon$ for the base student model. Then, we receive a final trajectory-level correctness reward with probability of $\epsilon^T$ assuming steps are independently completed. In contrast, by branching out from the expert trace at step $T - \tau$, STITCH only needs to complete the $\tau$ downstream steps, increasing the success to $\epsilon^\tau$. Notably, by increasing $\tau$ from 1 to $T$, STITCH facilitates curriculum learning by tackling individual problem steps one at a time requiring only $\Omega(1/\epsilon)$ samples to attain success per step.

*(3) If a question is hard, start by memorizing* RL-based training relies on preference or correctness rewards, which can be sparse and unstable. In contrast, SFT provides a strong fallback: the student memorizes canonical

traces, ensuring coherence and baseline ability even on hard questions. This aligns with cognitive intuition: humans first memorize worked examples before refining their skills through exploration. Thus, SFT is not merely an initialization step but an essential safeguard against RL instability.

*(4) Sample diverse trace to avoid SFT overfitting* A key risk of pure SFT is overfitting to a narrow distribution of traces, which can cause models to fail on semantically valid but stylistically different reasoning patterns. It is also known that an $L$-times deeper LLM can internally simulate $L$ chain-of-thought steps of a smaller LLM (Saunshi et al. (2025)). Consequently, expert models often generate long and dense traces that are prohibitively difficult for small student models to digest in one stage of supervised fine-tuning (SFT). Without proper strategy, the student may fail to align with the expert. Recent work (Li et al. (2025)) provides empirical evidence that small language models indeed struggle to learn directly from strong reasoners. To mitigate this, we deliberately sample traces from diverse expert models. We use sample traces from Llama3 (Grattafiori et al. (2024)), Qwen3 (Yang et al. (2025)) directly where as we rewrite sample traces from DeepSeek-R1 (Guo et al. (2025)) and GPT-OSS (Agarwal et al. (2025)) due to their longer trace generations. Thus we do not directly use the traces from DeepSeek-R1 or GPT-OSS but are rewritten with our planner model. Theoretically, this expands the student's support over reasoning space, reducing variance in downstream generalization error. Empirically, it ensures robustness to varied reasoning styles, thereby preventing over-specialization to a single teacher.

**Compressor**

*(1) Why we need summarization.* Transformer-based encoders (e.g., SBERT Reimers & Gurevych (2019), E5 Wang et al. (2022)) degrade when applied to very long spans. Summarization condenses essential semantics into a shorter sequence before encoding, yielding embeddings that better capture document-level meaning. Works like RAPTOR (Sarthi et al. (2024)) and MALRAG (Zheng et al. (2025)) show that hierarchical RAG benefits from summarized nodes: summaries act as "semantic bottlenecks," discarding redundancy while keeping high-level signals. Without summarization, embeddings of concatenated raw text tend to overweight surface-level lexical overlap rather than conceptual structure. Recent retrieval benchmarks (e.g., Kamradt (2024); Asai et al. (2024)) report that embeddings of summaries achieve higher recall and downstream QA accuracy than embeddings of raw large chunks, especially for multi-hop queries requiring abstraction.

*(2) Summarization changes the representation.* Summarization alters the text distribution, which in turn changes its embedding representation. As a result, embeddings of summaries are not equivalent to embeddings of the original content, motivating the need for a dedicated compression model.

## J    LIMITATION

SmartChunk's adaptive planner performs best on average, but for a certain type of datasets, some methods can indeed perform better. For example, GRAG shows strong performance on fact-based corpora such as QuALITY, where queries primarily require direct entity or fact retrieval rather than hierarchical reasoning.

