# OpenReview forum: "SmartChunk Retrieval: Query-Aware Chunk Compression with Planning for Efficient Document RAG"
_ICLR.cc/2026/Conference — ICLR 2026 Poster_

### Official Review · Reviewer_yPFE · 2025-10-27

**Soundness:** 3
**Presentation:** 2
**Contribution:** 3
**Rating:** 6
**Confidence:** 4

**Summary:**

This paper introduces SmartChunk, a retrieval-augmented generation (RAG) framework designed to address limitations of static chunking in long-document question answering. The core contributions include a lightweight planner that dynamically selects chunk sizes based on query complexity and document structure, a compression module that generates high-level chunk embeddings without expensive summarization, and STITCH, a reinforcement learning-based training method that combines RL and supervised fine-tuning to train the planner efficiently. The authors evaluate SmartChunk on five QA benchmarks and one out-of-domain dataset, demonstrating superior accuracy and efficiency compared to state-of-the-art baselines.

**Strengths:**

- The paper introduces a well-motivated approach to adaptive chunking in RAG systems.

- The paper thoughtfully addresses both accuracy and cost (monetary and latency), making the method highly relevant for real-world deployment.

**Weaknesses:**

- Several key implementation details appear to be omitted. Specifically:
    - The synthetic data pipeline lacks description of how chunks are merged or how the hierarchy is adjusted when initial answers are incorrect.
    - The test-time workflow is not explicitly outlined, leaving the inference process from query to final answer unclear.
- The fonts in Figure 2 are too small for comfortable reading. Furthermore, Figure 3a is challenging to interpret due to insufficient explanation of how the "performance gaps" are calculated and which specific "SOTA baselines" are being compared.

**Questions:**

Could you comment on the generalizability of SmartChunk to other RAG applications beyond QA?

---

> ### Author Response · Authors · 2025-11-20
>
> Thank you for the encouraging review. We appreciate your recognition of our methodology and the thoughtful suggestions. Below, we provide detailed responses to your questions and concerns.
>
> **W1: implementation details**
>
> **How chunks are merged during data synthesis**: During data synthesis, SmartChunk merges chunks using the same procedure as during inference. Specifically, higher-level embeddings are generated through the compression module, ensuring consistency between training and inference. Thank you for your suggestion, we clarified in the manuscript.
>
> **How the hierarchy is adjusted when initial answers are incorrect**: As described in Lines 281–282, if the initial answer is incorrect, the system expands retrieval (i.e., increases k) by incorporating additional chunks, enabling the model to revisit and potentially recover the correct answer. If the model still fails to solve the question even after retrieving all possible chunks, the sample is skipped.
>
>
> **Test-time workflow is not explicitly outlined**: We explain the Overview of SmartChunk Retrieval in Section 4.1 and Figure 2 (left) visualizes the test-time workflow. Following your suggestion, we have also added pseudo-code (See Algorithm 1 in revised manuscript) for the test-time pipeline in the Appendix. In addition, Figure 8 in the Appendix illustrates the differences between the test-time workflow of our SmartChunk retrieval and the vanilla RAG pipeline.
>
> **W2: Figure 3a is challenging to interpret**
>
> We appreciate the reviewer’s comment and have clarified the explanation of Figure 3a in the revised manuscript. Specifically, the “performance gaps” refer to the accuracy improvement of SmartChunk over each state-of-the-art (SOTA) baseline.
> $\\Delta_{\\text{gap}}=\\text{Performance(SmartChunk)}-\\text{Performance(Baseline)}$
> where the performance are evaluated with ROUGE, F1 or accuracy, as described in Figure 3 caption. The SOTA baselines compared include RAPTOR, MAL-RAG, and GRAG, as detailed in the legend of Figure 3a.
>
> **Q1: the generalizability of SmartChunk to other RAG applications beyond QA**
>
> Thank you for the suggestion. We conduct experiments to further demonstrate SmartChunk’s generalizability to RAG applications beyond QA. In particular, we evaluate SmartChunk on the summarization task on CORD-19 dataset[1], a new domain (medical) that is not represented in our training data, and a new RAG task (summarization) that differs significantly from QA. Even without any in-domain fine-tuning, and using only three in-context examples for few-shot prompting, the planner trained on other datasets substantially outperforms a fixed-size chunking baseline in both ROUGE-1 and ROUGE-2 (higher is better), while maintaining low monetary cost and latency, and matches the performance of MAL RAG, while using much lower monetary cost and latency. The results are shown below. These findings highlight that SmartChunk Retrieval naturally and effectively extends to non-QA RAG tasks such as summarization.
>
>
> |Method|ROUGE-1|ROUGE-2|Monetary Cost|Latency|
> |-|-|-|-|-|
> Fixed-size chunking | 36.61| 13.89 | 0.008 | 1.04 |
> MAL RAG | 44.79| 18.49 | 0.277 | 4.42 |
> SMARTCHUNK + few shot | 44.84| 18.53 | 0.060| 4.39 |
>
>
> [1] Wang, Lucy Lu, et al. "Cord-19: The covid-19 open research dataset." Proceedings of the 1st Workshop on NLP for COVID-19 at ACL 2020. 2020.

---

> ### Author Response · Authors · 2025-11-24
> **Follow up on the reviews**
>
> We again thank you for the detailed and constructive feedback. Hope our responses have clarified and answered all the points. In that case, we would love to know if the score can be increased. Or we would be happy to clarify if there is any other follow-up answer required?

---

### Official Review · Reviewer_iEV9 · 2025-10-28

**Soundness:** 3
**Presentation:** 3
**Contribution:** 3
**Rating:** 6
**Confidence:** 3

**Summary:**

This paper presents SmartChunk, a query-adaptive framework for retrieval-augmented generation (RAG) that addresses the limitations of static chunking strategies in long-document question answering. The key innovation lies in dynamically selecting optimal chunk granularity based on query characteristics rather than using fixed-size chunks.

The framework introduces three main components: (1) A planner that predicts the smallest and largest chunk sizes needed to answer a query, trained using a novel reinforcement learning approach called STITCH (Solve with RL, Then Imitate To Close Holes); (2) A chunk compression encoder that produces high-level embeddings without expensive LLM-based summarization; and (3) A multi-level chunking hierarchy that balances fine-grained detail with computational efficiency.

The authors evaluate SmartChunk on five QA benchmarks (NarrativeQA, QASPER, QuALITY, Natural Questions, and NewsQA) and demonstrate consistent improvements over state-of-the-art RAG baselines, achieving higher accuracy while reducing monetary costs by approximately 30%. The STITCH training methodology combines reinforcement learning with supervised fine-tuning in a stable loop, addressing challenges of noisy pseudo-labels and multi-objective optimization.

This paper presents a solid contribution to the RAG literature with a practical solution to an important problem. The query-adaptive chunking approach is intuitive and well-executed, with comprehensive experimental validation. The STITCH training methodology, while complex, addresses real challenges in this domain.

However, the contribution is somewhat incremental, and the added complexity may limit practical adoption. The improvements, while consistent, are not dramatic enough to represent a major breakthrough. The work would benefit from stronger theoretical foundations and more analysis of limitations.

The paper is above the acceptance threshold due to its practical relevance, solid experimental work, and novel training methodology, but it falls short of being a strong accept due to the incremental nature of the contribution and complexity concerns.

**Strengths:**

1. Practical relevance: Addresses a real bottleneck in current RAG systems where fixed chunking strategies perform poorly across diverse queries and documents.
2. Technical innovation: The STITCH training methodology is novel and addresses genuine challenges in training planners with noisy pseudo-labels and multi-objective rewards.
3. Comprehensive evaluation: Thorough experimental validation across multiple datasets with different characteristics, including out-of-domain evaluation.
4. Efficiency gains: Demonstrates both accuracy improvements and cost reductions, which is crucial for practical deployment.
5. Ablation studies: Systematic analysis of each component's contribution validates the design choices.
6. Orthogonality: Shows that the approach can be combined with other RAG improvements for additional gains.

**Weaknesses:**

1. Complexity vs. gains: The STITCH training procedure adds significant complexity to achieve what appears to be modest improvements over simpler baselines. The cost-benefit trade-off may not justify the added complexity in all scenarios.
2. Limited theoretical analysis: While the empirical results are strong, the paper lacks theoretical analysis of when and why the approach should work better than alternatives.
3. Reproducibility concerns: The STITCH training involves multiple stages with various hyperparameters and design choices that may make reproduction challenging.
4. Scalability questions: The evaluation is limited to relatively small corpora. It's unclear how the approach scales to very large document collections or real-time applications.
5. Planner generalization: While out-of-domain results are promising, more analysis is needed on how the planner generalizes to truly novel domains or document types not seen during training.
6. Limited error analysis: The paper doesn't provide sufficient analysis of failure modes or cases where the adaptive chunking performs poorly.

**Questions:**

1. Training data requirements: How much training data is needed for the planner to achieve good performance? How does performance degrade with limited training data?
2. Computational overhead: What is the actual computational overhead of the planner during inference? How does this compare to the savings from more efficient chunking?
3. Hyperparameter sensitivity: How sensitive is the STITCH training procedure to hyperparameter choices? Are there guidelines for setting these parameters for new domains?
4. Failure mode analysis: Can you provide more analysis of when the adaptive chunking fails? Are there query types or document structures where fixed chunking performs better?
5. Real-world deployment: Have you tested this approach in production settings? What are the practical challenges in deploying the full pipeline?
6. Comparison with simpler alternatives: How does the approach compare to simpler adaptive strategies, such as using query length or complexity as heuristics for chunk size selection?

---

> ### Author Response · Authors · 2025-11-20
> **Part 1**
>
> Thank you for your positive feedback on the manuscript. We appreciate your recognition of the approach and constructive feedback. Here, we address the questions and concerns.
>
> **W1: Complexity vs. gains**
>
> Thank you for pointing this out. While our method introduces additional training cost, we emphasize that this cost is incurred only once. The learned planner and compression encoder are universal components that can be directly applied across different datasets and domains, and we further demonstrate strong generalization to out-of-distribution data. In realistic RAG deployments, where the system serves millions of user queries, the one-time training cost is minor compared to the continuously accumulating inference cost of baseline methods(more than 5 times compare to ours). Also, we only fine-tune a lightweight 1.5B-parameter model (Qwen2.5-1.5B-Instruct), while the generator and summarizer in the RAG pipeline is a much larger GPT-class model whose inference dominates the overall cost. To contextualize the difference, the FLOPs of a single inference of GPT-4 (~175 B) roughly equals about 40 training steps of Qwen-1.5B on the same sequence length[1]. Moreover, our STITCH design mitigates the training overhead by requiring substantially fewer labeled examples and training steps. In the revised manuscript, we add Figure 3 to visualize the total end-to-end cost of SmartChunk and MAL RAG as the number of questions increases. The total cost including both the training(data collection + training cost of compressor and planner) and test-time cost. While SmartChunk incurs a small fixed cost upfront due to planner and encoder training, its test-time cost grows very slowly. In contrast, MAL RAG begins at zero but scales fasr with the number of questions, surpassing SmartChunk at around 2000 queries.
>
> **W2: Theoretical analysis**
>
> While the primary contribution of this work is algorithmic rather than theoretical, we do include supporting theoretical analysis in Appendix I: Design Choices and Justification. This analysis provides conceptual grounding for key components of our approach. For instance, we provide a theoretical explanation for STITCH’s curriculum learning effect. We also justify why the planner should be fine-tuned as a reasoning model rather than treated as a simple classifier. Specifically, introducing intermediate latent variables is shown to reduce sample complexity and improve generalization under non-linear mappings [2,3], suggesting that reasoning traces act as scaffolding to stabilize learning and enhance robustness to distribution shift.
>
> **W3: Reproducibility concerns**
>
> We appreciate the reviewer’s concern regarding reproducibility. We would like to clarify that STITCH introduces only one additional hyperparameter beyond those used in vanilla GRPO, the number of failed samples used to trigger the SFT phase. We add ablation experiments showing that STITCH’s performance is not sensitive to this hyperparameter in the revised manuscript. Using 64 failed samples (half of the default 128) yields comparable results, as shown in Table 8, with accuracies of 0.366 vs. 0.367. All other hyperparameters, such as learning rate and number of rollouts, follow the standard configurations of vanilla GRPO, ensuring that STITCH remains straightforward to reproduce.
>
>
> Moreover, we have demonstrated that STITCH generalizes robustly across tasks, such as mathematical reasoning, using the same hyperparameter settings. This indicates that the method is stable and easily reproducible.
>
> [1] Snell, Charlie, et al. "Scaling llm test-time compute optimally can be more effective than scaling model parameters." arXiv preprint arXiv:2408.03314 (2024).
>
> [2] Kevin Bello and Jean Honorio. Learning latent variable structured prediction models with gaussian perturbations. Advances in Neural Information Processing Systems, 31, 2018.
>
> [3] Shaojie Li and Yong Liu. Towards sharper generalization bounds for structured prediction. Advances in Neural Information Processing Systems, 34:26844–26857, 2021.

---

> ### Author Response · Authors · 2025-11-20
> **Part 2**
>
> **W4: Scalability questions.**
>
>
> We conduct experiments across datasets of varying scales, ranging from 5k to 62k document lengths, and observe consistent improvement. While our current evaluation focuses on moderate-sized corpora for controlled comparison with prior RAG baselines, SmartChunk is inherently scalable because the planner operates at the chunk-level index rather than the token level. This design allows efficient retrieval decisions even when the underlying corpus grows by orders of magnitude.
>
>
>
>
> In practice, SmartChunk can be seamlessly integrated with shared vector databases (e.g., FAISS [4], ScaNN [5]) and supports parallel retrieval across chunk hierarchies, enabling real-time performance. The adaptive planner also reduces the number of retrieval calls per query, lowering cost and latency as corpus size increases.
>
> **W5: Planner generalization**
>
> The NewsQA dataset used for our out-of-domain generalization experiments represents a substantially different domain from in-domain datasets such as NarrativeQA. While NarrativeQA features long-form, story-driven passages that require sequential reasoning and discourse-level comprehension, NewsQA consists of factual, entity-centric news articles with minimal narrative structure. The planner’s strong performance on both datasets demonstrates its robustness to variations in content style, reasoning type, and document organization.
>
>
> The generalization arises because the planner relies on semantic and structural cues (e.g., question type, discourse depth, metadata) rather than domain-specific patterns. So we believe the planner can generalize to truly novel domains or document types not seen during training.
>
>
> Furthermore, we will conduct additional experiments to demonstrate SmartChunk’s generalizability to other RAG applications beyond QA. We conduct experiments to further demonstrate SmartChunk’s generalizability to RAG applications beyond QA. In particular, we evaluate SmartChunk on the summarization task on CORD-19 dataset[6], a new domain (medical) that is not represented in our training data, and a new RAG task (summarization) that differs significantly from QA. Even without any in-domain fine-tuning, and using only three in-context examples for few-shot prompting, the planner trained on other datasets substantially outperforms a fixed-size chunking baseline in both ROUGE-1 and ROUGE-2 (higher is better), while maintaining low monetary cost and latency, and matches the performance of MAL RAG, while using much lower monetary cost and latency. The results are shown below. These findings highlight that SmartChunk Retrieval naturally and effectively extends to non-QA RAG tasks such as summarization.
>
> |Method|ROUGE-1|ROUGE-2|Monetary Cost|Latency|
> |-|-|-|-|-|
> Fixed-size chunking | 36.61| 13.89 | 0.008 | 1.04 |
> MAL RAG | 44.79| 18.49 | 0.277 | 4.42 |
> SMARTCHUNK + few shot | 44.84| 18.53 | 0.060| 4.39 |
>
> **W6: Limited error analysis**
>
> As discussed in section 5.3, SmartChunk’s adaptive planner performs best on average, but for a certain type of datasets, some methods can indeed perform better. For example, GRAG shows strong performance on knowledge-centric dataset such as QuALITY, where queries primarily require direct entity or fact retrieval rather than hierarchical reasoning. The results are shown in Figure 4(a) and the exact numbers can be found in Table 9. Thank you for your suggestion. We include an additional Limitation section in the revised manuscript.
>
> **Q1: Training data requirements**
>
> Our training pipeline is highly sample-efficient. As shown in Table 5, STITCH achieves higher QA accuracy (0.564 vs. 0.538) compared to the vanilla SFT+RL pipeline while using about half the amount of labeled data. We further analyze data efficiency in Figure 5. Following the standard GRPO setup, we train for 200 steps, but observe that performance stabilizes around step 100, indicating that fewer than 25k samples are sufficient for convergence.
>
>
>
>
>
> [4] Matthijs Douze, Alexandr Guzhva, Chengqi Deng, Jeff Johnson, Gergely Szilvasy, Pierre Emmanuel Mazar´e, Maria Lomeli, Lucas Hosseini, and Herv´e J´egou. The faiss library. 2024.
>
>
> [5] Philip Sun, David Simcha, Dave Dopson, Ruiqi Guo, and Sanjiv Kumar. Soar: Improved indexing for approximate nearest neighbor search. In Neural Information Processing Systems, 2023b. URL https://arxiv.org/abs/2404.00774.
>
> [6] Wang, Lucy Lu, et al. "Cord-19: The covid-19 open research dataset." Proceedings of the 1st Workshop on NLP for COVID-19 at ACL 2020. 2020.

---

> ### Author Response · Authors · 2025-11-20
> **Part 3**
>
> **Q3: Hyperparameter sensitivity**
>
> See W3. STITCH introduces only one additional hyperparameter beyond those used in vanilla GRPO. We add ablation experiments showing that STITCH’s performance is not sensitive to this hyperparameter in the revised manuscript.
>
>
> **Q4: Failure mode analysis**
>
> See W6. As discussed in section 5.3, SmartChunk’s adaptive planner performs best on average, but for a certain type of datasets, some methods can indeed perform better.
>
> **Q5: Real-world deployment**
>
> We have tested and deployed SmartChunk methodology in stage setting which is generally considered as the replica of the real world settings. Being smaller in size the planner model can be hosted in 24GB A10G and with further quantization we are able to use it in smaller GPUs. It consumes less memory during the inference. We have hosted both in in-network and distributed settings. The full retrieval pipeline is very much flexible and can accommodate any LLM as a generator (including Blackbox model APIs). So in practical settings wherever the vector database RAG with end generator LLM is hosted, with the inclusion of a single GPU we are able to host the full SmartCHunk Retrieval pipeline.
>
> **Q6: Comparison with simpler alternatives**
>
> We appreciate the reviewer’s question. To examine whether simpler heuristics could replace SmartChunk, we analyzed the correlation between query length and the planner’s chosen chunk level. The Pearson correlation coefficient is 0.07, indicating no meaningful relationship. And using query length as heuristics for chunk size selection can only achieve 0.369 QA accuracy, which is much lower than SmartChunk’s 0.564 accuracy.
>
>
> In contrast, SmartChunk’s query-aware planner captures deeper semantic and reasoning cues, enabling more accurate and efficient chunk selection.

---

> ### Author Response · Authors · 2025-11-24
> **Follow up on the reviews**
>
> We again thank you for the detailed and constructive feedback. Hope our responses have clarified and answered all the points. In that case, we would love to know if the score can be increased. Or we would be happy to clarify if there is any other follow-up answer required?

---

### Official Review · Reviewer_zCxf · 2025-10-30

**Soundness:** 1
**Presentation:** 2
**Contribution:** 2
**Rating:** 4
**Confidence:** 4

**Summary:**

The paper aims to address the trade-off dilemma between accuracy and cost in existing RAG systems. SmartChunk introduces two modules: The planner dynamically predicts the minimum and maximum chunk levels required to answer a user query upon receiving it. The chunk compression encoder generates high-level summary embeddings from embeddings of low-level chunks. The paper also proposes a sophisticated hybrid training scheme.

**Strengths:**

1. The paper identifies the most significant pain point of current advanced RAG systems: high costs.
2. In response to the exorbitant costs of LLM-based summarization, the paper proposes the chunk compression encoder.
3. The paper implements a dynamic RAG system, claiming to achieve a trade-off between accuracy and cost.

**Weaknesses:**

1. The motivation behind the planner is extremely difficult to comprehend. The core assumption of the paper is that a query-based planner can predict, prior to retrieval, the minimum and maximum chunk levels required to answer a question. However, given that the information distribution within documents is unknown, such predictions lack sufficient informational support and exhibit low scientific rigor.
2. Why must the information required to answer a question be precisely distributed across a continuous range of chunk levels? For instance, a user may need a document summary to grasp the global context while simultaneously requiring a few precise sentences to extract key facts. The rigid interval design proposed in the paper is disconnected from the nonlinear information needs encountered in real-world scenarios.
3. The paper attempts to use a query-aware model to plan a problem that should ideally be decided based on document structure during indexing. Although the paper claims to be selecting from pre-existing hierarchical layers, this constitutes a guess made on the basis of severely inadequate information.
4. STITCH combines SFT, RL, Prompt-based RL, and Imitation Learning. This complexity exposes the poorly defined nature of the planner's task itself. If SFT + RL is effective, why is imitation learning still necessary? The paper employs extremely high training complexity and engineering techniques to forcibly fit a task with questionable motivation.
5. Reward signals in reinforcement learning are challenging to define. The RL reward is derived from the final QA accuracy, which is an extremely sparse and significantly delayed signal. The planner predicts intervals, the retriever fetches relevant information, and the generator determines answer correctness. RL simply attributes the final error to the initial planner, which is methodologically untenable.

**Questions:**

1. Is such extremely high training complexity a necessary condition for achieving adaptive planning, or is it a design choice made to compensate for deficiencies?
2. Given the extremely long, multi-stage execution chain from planning to the final answer, how do the authors address the long-range credit assignment problem?
3. Is the continuous interval assumption scientifically reasonable? In real-world scenarios, answering a complex question may require a non-continuous, cross-granularity combination of information. Could this rigid interval prediction paradigm become a fundamental bottleneck for the system in handling multi-hop or complex reasoning tasks?

---

> ### Author Response · Authors · 2025-11-20
> **Part 1**
>
> Thank you for your detailed and constructive feedback. Below, we address the reviewer’s comments point by point. We would also be grateful to hear any additional feedback or questions.
>
> **W1: Given that the information distribution within documents is unknown, the query-based planner lack sufficient informational support.**
>
> We would like to clarify that the planner is not based solely on the query. It also sees document metadata (e.g., section titles, source domains, and even a small part of the document content), allowing it to exploit predictable patterns.
>
>
> We also show concrete examples of planner inputs and outputs in Appendix E which we believe addresses the reviewer’s concern. These examples demonstrate that the planner is not making arbitrary or unsupported guesses about chunk levels. Instead, it leverages query semantics and document metadata, to think and make decisions. We do see some evidence from the generated output. In addition to the examples shown in Appendix E, we present further examples of the planner generated output below::
>
> *"The query targets a specific experimental setting, which is conventionally localized in the experiment section; accordingly,  assigns the section level as the maximum chunk size."*
>
> *"The document corresponds to narrative fiction and entails long-range temporal dependencies; therefore, the generator favors larger units, such as paragraph-level chunks."*
>
>
> Prior work [1–4] has shown that different question types and document structures benefit from different granularities. E.g., factoid questions typically require shorter, fine-grained chunks, whereas multi-hop or narrative queries benefit from larger ones. This regularity, also reflected in our experiments (Figure 4b, 5 in revised manuscript), provides sufficient informational support for the planner’s decisions.
>
> **W2: The rigid interval design proposed in the paper is disconnected from the nonlinear information needs encountered in real-world scenarios.**
>
> The scenario raised by the reviewer, requiring both broad global summaries and fine-grained factual details, is precisely the motivation behind SmartChunk. Our system is designed to provide the generator with a cross-granularity combination of information, for example providing both document-level context and sentence-level evidence for a single query when needed.
>
> For example, as shown in Appendix F, the planner predicts the largest and smallest chunk sizes needed for a given query–document pair (e.g., {“largest chunk size”: “section”, “smallest chunk size”: “sentence”}). All chunk levels within this interval are then provided to the retriever and generator. In this example, sentence-level, multi-sentence-level, paragraph-level, and section-level chunks are all included in the multi-level hierarchy, ensuring that both global summaries and fine-grained facts are available. The retriever then selects the most relevant chunks from this multi-granularity pool. Thus, in the scenario mentioned by the reviewer, SmartChunk naturally retrieves both the broader contextual summary and the key precise sentences, providing the generator with the complete information needed to answer the query effectively. In contrast, existing RAG methods rely on fixed, rigid chunk sizes, making them poorly suited for such information needs.
>
> Our experiments (Figure 4b, 5) demonstrate that SmartChunk’s design is inherently flexible rather than rigid. For example, we visualize the distribution, in Figure 5. The adaptive behavior also manifests in the planner’s predicted range of chunk levels. In NarrativeQA, both the minimum and maximum chunk levels are distributed toward the higher end (levels 4–6), reflecting the need for broader contextual integration to capture long-range narrative dependencies. In contrast, QuALITY exhibits a skew toward smaller chunk levels (1–3), as its questions typically target specific facts or short reasoning chains within individual paragraphs.
>
> [1] Parth Sarthi, Salman Abdullah, Aditi Tuli, Shubh Khanna, Anna Goldie, and Christopher D Manning. Raptor: Recursive abstractive processing for tree-organized retrieval. In The Twelfth International Conference on Learning Representations, 2024.
>
> [2] Zheng Zheng, Xinyi Ni, and Pengyu Hong. Multiple abstraction level retrieve augment generation. arXiv preprint arXiv:2501.16952, 2025.
>
> [3] Sinchana Ramakanth Bhat, Max Rudat, Jannis Spiekermann, and Nicolas Flores-Herr. Rethinking chunk size for long-document retrieval: A multi-dataset analysis. arXiv preprint arXiv:2505.21700, 2025.
>
> [4] Michael Gunther, Isabelle Mohr, Daniel James Williams, Bo Wang, and Han Xiao. Late chunking: contextual chunk embeddings using long-context embedding models. arXiv preprint arXiv:2409.04701, 2024.

---

> ### Author Response · Authors · 2025-11-20
> **Part 2**
>
> **W3: Using  document structure**
>
>
> We thank the reviewer for raising this important point. We would like to clarify that SmartChunk is designed to complement, not replace, structure-based indexing. Structural information can be seamlessly incorporated into our framework.
>
>
> We agree that ideally, there can be a better design. But considering real application, there are some challenges. In real-world applications, many documents, such as news articles, reports, or web pages, lack consistent or rich hierarchical structure, whereas our sentence- and paragraph-level segmentation provides a more general and robust foundation. Also, relying solely on document structure during indexing is often insufficient, as the optimal retrieval granularity depends strongly on the query intent.
>
>
> Most importantly, among recent state-of-the-art retrieval methods, SmartChunk consistently achieves higher accuracy and efficiency, demonstrating the effectiveness of our SmartChunk Retrieval.
>
> **W4: If SFT + RL is effective, why is imitation learning still necessary?**
>
>
> We clarify the motivation for introducing a fine-tuned planner in the response to W1-2, and we hope this addresses the concern.
>
>
> Traditional SFT + RL is neither efficient nor effective. As shown in Table 5, STITCH achieves about 6% higher planning accuracy and 3% final QA accuracy with half of labeled data. Traditional SFT + RL approaches require large amounts of high quality labeled data and careful handcrafted hyperparameter tuning. As prior work shows[5-8], small models often fail to learn complex traces generated by large models, and training length is delicate: too few SFT epochs lead to sparse-reward failure in RL, while excessive fine-tuning causes overfitting and degraded RL. As you mentioned in W5 there is also a sparse reward challenge for vanilla RL. These dependencies introduce a significant engineering burden. In contrast, STITCH automatically bridges SFT and RL through curriculum learning. STITCh targeting the sparse reward issue by introducing branched rollout.  As shown in Table 5, STITCH achieves the highest-performing planner while using substantially fewer labeled examples.
>
>
> Appendix F further demonstrates strong generalization to other domains such as mathematical reasoning, where SFT + RL struggles on harder instances. We provide more justifications in Appendix I.
>
> **W5: Reward signals in reinforcement learning are challenging to define.**
>
> We agree that defining reward signals is inherently challenging. Indeed, this challenge is precisely what motivates the design of our STITCH framework and directly addresses the reviewer’s question in W4 about why STITCH is needed. Unlike standard RL approaches that rely solely on final QA accuracy as a sparse, end-of-episode reward, STITCH introduces branched rollouts and intermediate feedback to provide denser and more informative supervision. When RL struggles on particularly difficult questions, we integrate imitation learning to guide the planner using successful trajectories, further mitigating delayed credit assignment and improving stability. We also provide theoretical analysis in Appendix I.
>
>
> Empirically, this hybrid design proves effective: STITCH consistently outperforms pure SFT+RL both in planner fine-tuning (Table 5) and in transfer to additional domains such as mathematical reasoning, particularly on challenging examples (Table 8, Figure 9).
>
> [5] Zhang, Xuechen, et al. "BREAD: Branched Rollouts from Expert Anchors Bridge SFT & RL for Reasoning." arXiv preprint arXiv:2506.17211 (2025).
>
> [6] Chu, Tianzhe, et al. "Sft memorizes, rl generalizes: A comparative study of foundation model post-training." arXiv preprint arXiv:2501.17161 (2025).
>
> [7] Zhang, Wenhao, et al. "On-policy rl meets off-policy experts: Harmonizing supervised fine-tuning and reinforcement learning via dynamic weighting." arXiv preprint arXiv:2508.11408 (2025).
>
> [8] On-Policy Distillation https://thinkingmachines.ai/blog/on-policy-distillation/

---

> ### Author Response · Authors · 2025-11-20
> **Part 3**
>
> **Q1: Is such extremely high training complexity a necessary condition for achieving adaptive planning, or is it a design choice made to compensate for deficiencies?**
>
> As explained in the response to W4, STITCH actually needs lower training complexity and avoids the significant engineering burden of vanilla SFT+RL.
>
> **Q2: Given the extremely long, multi-stage execution chain from planning to the final answer, how do the authors address the long-range credit assignment problem?**
>
> As explained in the response to W5, STITCH introduces branched rollouts and intermediate feedback to provide denser and more informative supervision. When RL struggles on particularly difficult questions, we integrate imitation learning to guide the planner using successful trajectories, further mitigating delayed credit assignment and improving stability.
>
> **Q3: Is the continuous interval assumption scientifically reasonable? In real-world scenarios, answering a complex question may require a non-continuous, cross-granularity combination of information. Could this rigid interval prediction paradigm become a fundamental bottleneck for the system in handling multi-hop or complex reasoning tasks?**
>
> As explained in the response to W2, SmartChunk retrieval is more flexible and adaptive than current RAG pipeline and sota papers. It can provide a cross-granularity combination of information.

---

> ### Author Response · Authors · 2025-11-24
> **Follow up on the reviews**
>
> We again thank you for the detailed and constructive feedback.
> Hope our responses have clarified and answered all the points. In that case, we would love to know if the score can be increased.
> Or we would be happy to clarify if there is any other follow-up answer required?

---

> ### Comment · Reviewer_zCxf · 2025-11-26
>
> Thank you for serious clarifications provided. However, I still have several concerns regarding W1, W2, and W3, as outlined below.
>
> Regarding W1: In the current experimental setup, the paper does not provide a quantitative analysis of how different types of input information contribute to the planner's performance. Since the authors claim that the planner can make better decisions by leveraging“ the query + document metadata + small content snippets,” it would be important to compare at least the following variants:
> * planner based only on the query;
> * query + document metadata;
> * query + the full document content.
>
> Without such ablation studies, it is difficult for readers to understand why the planner is able to make accurate predictions.
>
> Regarding W2 and W3: In the rebuttal, the authors again state that “existing RAG methods rely on fixed, rigid chunk sizes”. This characterization does not accurately reflect the current state of the RAG literature and obscures the genuine incremental contribution of SmartChunk beyond existing dynamic / hierarchical / planner-based approaches.
>
> Chunking is a central focus of the paper, yet the experiments only compare against the most basic fixed-size chunking strategy, without including semantic chunking or agent-based chunking baselines [1-3]. Moreover, Table 2 reports main results on only a single dataset, and no new experiments were added in the rebuttal, so the concerns are not convincingly addressed.
>
> As a result, it remains possible that the apparent advantage of SmartChunk holds only relative to a clearly weaker baseline, which is precisely the source of my skepticism.
>
> [1] Duarte, André V., et al. "Lumberchunker: Long-form narrative document segmentation." Findings of the Association for Computational Linguistics: EMNLP 2024. 2024.
>
> [2] Jin, Jiajie, et al. "Hierarchical document refinement for long-context retrieval-augmented generation." Proceedings of the 63rd Annual Meeting of the Association for Computational Linguistics (Volume 1: Long Papers). 2025.
>
> [3] Ni, Tongke, et al. "CrossFormer: Cross-Segment Semantic Fusion for Document Segmentation." arXiv preprint arXiv:2503.23671 (2025).

---

> ### Author Response · Authors · 2025-12-01
>
> **We are happy to hear that W4, W5, and all other questions have been fully addressed.**
>
> Regarding **W1**, we appreciate the reviewer’s acknowledgement that the planner makes accurate predictions. As the reviewer suggested, we conduct additional experiments to further justify the effectiveness of our current setting.
>
> For **W2 and W3**, based on the comments, the reviewer recognizes that our method is flexible but believes we should compare against semantic-chunking or agent-based chunking baselines across multiple datasets. We would like to clarify that this comparison is **already included** in the paper (see Table 2, Figure 6 and Table 9). Reviewer DoNt, iEV9 and yPFE all mentioned that we evaluated SmartChunk on multiple benchmarks and compared it to state-of-the-art baselines. We further analyze and conduct experiments with the new baseline reviewer suggested and find that our method still clearly outperforms.
>
> We provide detailed explanations and additional experiments below.
>
> Regarding W1, as we explained in the rebuttal, the planner leverages both query features and document metadata to reason and make decisions. These signals already provide sufficient information for the planner to accurately predict the appropriate chunk range.  We further support this claim with new experiments following your suggestion. Specifically, we evaluated planning accuracy under different input configurations. Our setting, which includes both the query and document metadata, outperforms all other variants. When only the query is provided, performance drops significantly because the planner lacks document information. When we use the full document content, Qwen2.5 models have a context limit of 32,768 tokens, which is shorter than many documents in our dataset. This makes it infeasible to include the entire document for many samples. For fairness, we constrained all inputs to be shorter than 32k tokens, cutting documents when necessary. Even under this controlled setup, we find that LLMs often struggle with extremely long contexts, and as a result, the full-content setting performs worse than the metadata-based approach.
>
> |Method|Planning accuracy|
> |-|-|
> Ours | 0.820 |
> Query only | 0.786 |
> Query +  the full document content | 0.802 |
>
> Regarding W2 and W3, we evaluate SmartChunk on five QA benchmarks(NarrativeQA, QASPER, QuALITY, Natural Questions, and NewsQA) including one out-of-domain dataset NewsQA. Table 1 lists dataset statistics used in our evaluation. Table 2 shows the average performance and Table 9 shows the performance for each dataset separately. We conduct experiments with SOTA baselines instead of only basic fixed-size chunking strategies. As shown in Table 2, we compare to MAL RAG leverages chunks at multiple abstraction levels, GRAG integrates graph-based representation, and  RAPTOR builds a multi-level chunk tree by recursively embedding, clustering, and summarizing text from the bottom up, and performs retrieval over this hierarchical structure. In Figure 6, we also demonstrate that SMARTCHUNK is orthogonal to recent advanced chunking strategies. As for the papers mentioned by the reviewer, [1,3] is a semantic chunking, as shown in figure 6, SMARTCHUNK is orthogonal to semantic chunking methods. Integrating SMARTCHUNK with these methods
> gets further gains in QA accuracy. Also, as they leverage an LLM to segment documents, it’s pretty expensive. For [1], only segment documents take  \$0.294 which is much more expensive than the whole SmarkChunk which takes 0.078. LongRefiner [2] refine long retrieved documents before LLM processing. For fair comparison, we conduct experiments on NewsQA which is out-of-domain for both ours and LongRefiner. LongRefiner achieves slightly better performance 0.3% but takes much more monetary cost(0.032 vs 0.058) and latency(1.34 vs 2.40). Our SmartChunk avoids running a heavy refiner LLM online and lets us plug into standard embedding-based retrievers and generators. We also propose a chunk embedding compressor that additionally reduces the cost.
>
> |Method| F1| Monetary cost($)| Latency (s)|
> |-|-|-|-|
> |SMARTCHUNK| 0.875| 0.026| 1.29|
> |SMARTCHUNK + few shot| 0.906 | 0.032| 1.34|
> |LongRefiner| 0.909 | 0.058| 2.40|

---

### Official Review · Reviewer_DoNt · 2025-11-11

**Soundness:** 2
**Presentation:** 3
**Contribution:** 3
**Rating:** 6
**Confidence:** 3

**Summary:**

The paper presents SmartChunk, a RAG framework that overcomes the limitations of static chunking. The method has two components, a Planner that predicts the chunk granularity for a given query, and a Chunk Compression Encoder that generates high-level embeddings directly from lower-level chunks without summarization. To train the planner without ground-truth labels, the authors introduce STITCH, a training method combining RL, expert hints, and imitation learning. Experiments across five benchmarks show SmartChunk having comparable performance with strong baseline while reducing costs.

**Strengths:**

The paper is well written and the method is well ablated with each part showing the tradeoffs. The method has comparable performance to methods like RAPTOR, GRAG, MAL RAG. The Chunk Compression Encoder is specfically interesting, It is a surprising result how it can boost results compared to directly just embedding the document. The method also shows some generalization.

**Weaknesses:**

Compared to some previous methods like RAPTOR, SmartChunk requires training, for the planner and the compression encoder. A few important baselines that are currently mssing is just retrieving from the database with differing chunk levels i.e. the model can retireve from all the chunks together (at different token levels) where the tokens can be embedded normally and also via the chunk compression encoder.

**Questions:**

Beyond the average length of the retrieved documents, what is the distrubution of actaully the levels being chosen both across datasets and within the same dataset?
How does the method compare to other methods in terms of training efficiency and time?
How would the model compare to the baselines mentioned in the weaknesses section?

---

> ### Author Response · Authors · 2025-11-20
>
> Thank you for the detailed and constructive feedback. Here, we answer the questions and address the concerns.
>
> **W1: SmartChunk requires training for the planner and the compression encoder.**
>
> Thank you for pointing this out. While our method introduces additional training cost, we emphasize that this cost is incurred only once. The learned planner and compression encoder are universal components that can be directly applied across different datasets and domains, and we further demonstrate strong generalization to out-of-distribution data. In realistic RAG deployments, where the system serves millions of user queries, the one-time training cost is minor compared to the continuously accumulating inference cost of baseline methods(more than 5 times compare to ours). Also, we only fine-tune a lightweight 1.5B-parameter model (Qwen2.5-1.5B-Instruct), while the generator and summarizer in the RAG pipeline is a much larger GPT-class model whose inference dominates the overall cost. To contextualize the difference, the FLOPs of a single inference of GPT-4o roughly equals about 40 training steps of Qwen-1.5B on the same sequence length[1]. Moreover, our STITCH design mitigates the training overhead by requiring substantially fewer labeled examples and training steps. In the revised manuscript, we add Figure 3 to visualize the total end-to-end cost of SmartChunk and MAL RAG as the number of questions increases. The total cost including both the training(data collection + training cost of compressor and planner) and test-time cost. While SmartChunk incurs a small fixed cost upfront due to planner and encoder training, its test-time cost grows very slowly. In contrast, MAL RAG begins at zero but scales fast with the number of questions, surpassing SmartChunk at around 2000 queries.
>
> **W2: A few important baselines that are currently missing.**
>
> Thank you for the helpful suggestion. We note that this baseline is already included in our paper (Table 2, Ours w/o p), same as what you suggested, the model retrieves from all the chunks together (at different token levels), the tokens are embedded with the chunk compression encoder. Compared to our SmartChunk retrieval, this static multi-granular retrieval leads to lower QA accuracy and higher inference cost. In addition, retrieving from all chunks with normal embeddings corresponds to the MAL-RAG baseline, which incurs higher monetary cost(~5×) and latency while still performing worse in accuracy. These comparisons confirm that SmartChunk’s advantage comes from adaptive planning and compression encoder rather than simply accessing all chunk granularities. We provide detailed analysis in Section 5.
>
> **Q1: Beyond the average length of the retrieved documents, what is the distribution of actually the levels being chosen both across datasets and within the same dataset?**
>
> Thank you for the suggestion. We add a figure to visualize the distribution, see Figure 5 in the revised manuscript. The adaptive behavior also manifests in the planner’s predicted range of chunk levels. In NarrativeQA, both the minimum and maximum chunk levels are distributed toward the higher end (levels 4–6), reflecting the need for broader contextual integration to capture long-range narrative dependencies. In contrast, QuALITY exhibits a skew toward smaller chunk levels (1–3), as its questions typically target specific facts or short reasoning chains within individual paragraphs.
>
> **Q2:  How does the method compare to other methods in terms of training efficiency and time?**
>
> In section 5.3, we discuss the efficiency of STITCH. The results are shown in Table 5. We also visualize the training curve to provide better understanding of training cost and time, see Figure 7 in Appendix (Figure 5 in original version). In the revised manuscript, we also add Figure 3 to visualize the total end-to-end cost including both the training(data collection + training cost of compressor and planner) and test-time cost as the number of questions increases. The result shows that the training cost is negligible relative to the savings obtained through more efficient, query-adaptive chunking.
>
> **Q3: How would the model compare to the baselines mentioned in the weaknesses section?**
>
> We conduct experiments compared against standard and state-of-the-art RAG systems including Fixed-size chunking, Late chunking, MAL RAG, GRAG and RAPTOR. The detailed description of baselines are shown in section 5.1. As shown in Table 2 and 9,  SmartChunk consistently outperforms all baselines across key metrics.
>
>
> [1] Snell, Charlie, et al. "Scaling llm test-time compute optimally can be more effective than scaling model parameters." arXiv preprint arXiv:2408.03314 (2024).

---

> ### Author Response · Authors · 2025-11-24
> **Follow up on the reviews**
>
> We again thank you for the detailed and constructive feedback.
> Hope our responses have clarified and answered all the points. In that case, we would love to know if the score can be increased.
> Or we would be happy to clarify if there is any other follow-up answer required?

---

### Author Response · Authors · 2025-12-04
**Rebuttal summarization**

We are encouraged that the majority of reviewers provided positive assessments of the paper. We appreciate that many reviewers found our approach well-motivated, recognized the novelty of the proposed methods, and acknowledged the breadth of our state-of-the-art comparisons and ablation studies across diverse benchmarks. All reviewers acknowledge that our work tackles a key bottleneck in RAG systems, retrieval performance depends heavily on chunk size, often suffers from noisy or irrelevant evidence, and becomes inefficient at scale. To address this, we propose SmartChunk retrieval, a query-adaptive framework that uses (i) a planner that predicts the optimal chunk abstraction level for each query, and (ii) a lightweight compression module that produces high-level chunk embeddings without repeated summarization. The planner is trained using STITCH, a novel framework that bridges SFT and RL to overcome label scarcity, noisy supervision, and instability in multi-objective optimization. Reviewers DoNt, zCxf and iEV9 highlighted the novelty and practical relevance of this approach, noting its ability to adapt retrieval granularity while jointly optimizing accuracy and efficiency. Our experiments span five QA benchmarks and provide extensive ablations.

For weaknesses and questions, we provide detailed responses to each point and include additional experiments to further support our conclusions. We believe these clarifications and new results thoroughly address the concerns. '

## We summarize the common concerns and explain how our rebuttal resolves them.


**DoNt & iEV9: Complexity vs. gains.**

Beyond the inference cost we analyze in the paper, reviewer DoNt and iEV9 raised a concern that SmartChunk introduces additional training overhead, and that the training cost–benefit trade-off may not justify the added complexity in all scenarios.
**We address this concern in two ways:** **Clarification of cost:**  We emphasize that the planner’s training cost is incurred only once and is relatively small and include concrete numbers to illustrate this. The FLOPs of a single generator inference roughly equal about 40 training steps of planner/compressor.
**End-to-end comparison:** We added Figure 3, which visualizes the total end-to-end cost(training + inference). The baseline MAL RAG begins at zero but scales much faster with the number of questions, surpassing SmartChunk at around 2000 queries. The results further demonstrate that SmartChunk provides a favorable long-term cost–efficiency trade-off.

**iEV9, yPFE: the generalizability of SmartChunk to other RAG applications beyond QA.**

The reviewers already noted our out-of-domain results are promising and expressed interest in seeing evaluations on RAG tasks beyond QA. We **conduct new experiments** on the summarization task on a new domain (medical) and a new RAG task (summarization) CORD-19 dataset to further demonstrate SmartChunk’s generalizability. Without fine-tuning, the planner substantially outperforms fixed-size chunking (ROUGE 44.84 vs. 36.61), and matches the performance of SOTA baseline MAL RAG, while using much lower monetary cost (0.060 vs. 0.277) and latency(4.39 vs. 4.42).

## We believe we fully addressed the only negative reviewer zCxf's concern.
The W4, W5, and all other questions have been fully addressed after rebuttal**. We provide explanations and additional experiments to address remaining concerns in the second response.

**The remaining concerns of W2 and W3 appear to stem from a misunderstanding**, the reviewer states that we report main results on only a single dataset and compare only against a basic fixed-size chunking baseline, which is not true. We evaluated SmartChunk on five benchmarks and compared to SOTA baselines. We clarify our setting and **conduct additional experiments compared to the new baselines** the reviewer suggests. Compared to our SmartChunk Retrieval, LongRefiner achieves slightly better performance 0.3% but takes much more monetary cost(0.032 vs 0.058) and latency(1.34 vs 2.40).

**The original W1 also appears because of misunderstanding**. The reviewer thinks we only use queries as planner input, which is not true. After our clarification in rebuttal, the reviewer ask an additional ablation study. Following the reviewer’s suggestion, **we conduct a new ablation study** demonstrating that our design, where the planner leverages both query features and document metadata, provides sufficient information for accurately predicting the appropriate chunk range. When only the query is provided, planning accuracy drops significantly from 0.820 to 0.786 because the planner lacks document information. When whole documents are provided, LLMs still struggle with extremely long contexts, planning accuracy drops from 0.820 to 0.802. The experiments further justify the effectiveness of our current setting, so we believe the concern should be fully addressed.

---

### Meta-Review · Area_Chair_haeu · 2025-12-23

**Summary:**

1. Complexity vs. Gains: The method requires training for the planner and the compression encoder, which raises concerns about the cost-benefit trade-off. (Reviewer DoNt and iEV9)
2. Missing important baselines (e.g., with differing chunk levels). (Reviewer DoNt)
3. Concerns about the generalizability of SmartChunk to other RAG applications beyond QA. (Reviewer iEV9 and yPFE)
4. Missing some details: The synthetic data pipeline lacks description; The test-time workflow is not explicitly outlined, leaving the inference process from query to final answer unclear. (Reviewer yPFE)
5. Some figures are unclear. (Reviewer yPFE)

**Reviewer Concerns:**

The authors have addressed almost all the reviewers' questions and misunderstandings during the rebuttal phase. It is recommended to incorporate these discussions into the new version and further refine the paper to improve its quality.

**Reviewer Scores:**

Reviewer DoNt: 6 --> 8

Reviewer zCxf:  4 --> 6

Reviewer iEV9: retains 6

Reviewer yPFE:  retains 6

---

### Decision · Program_Chairs · 2026-01-26

Accept (Poster)